# Syntaxin-1A modulates vesicle fusion in mammalian neurons via juxtamembrane domain dependent palmitoylation of its transmembrane domain

Gülçin Vardar[1]*, Andrea Salazar-Lázaro[1], Sina Zobel[1], Thorsten Trimbuch[1], Christian Rosenmund[1,2]*

[1]Department of Neurophysiology, Charité - Universitätsmedizin Berlin, Humboldt-Universität zu Berlin, and Berlin Institute of Health, Berlin, Germany; [2]NeuroCure Excellence Cluster, Berlin, Germany

*For correspondence:
gulcinv@gmail.com (GV);
christian.rosenmund@charite.
de (CR)

Competing interest: The authors declare that no competing interests exist.

**Abstract** SNAREs are undoubtedly one of the core elements of synaptic transmission. Contrary to the well characterized function of their SNARE domains bringing the plasma and vesicular membranes together, the level of contribution of their juxtamembrane domain (JMD) and the transmembrane domain (TMD) to the vesicle fusion is still under debate. To elucidate this issue, we analyzed three groups of STX1A mutations in cultured mouse hippocampal neurons: (1) elongation of STX1A's JMD by three amino acid insertions in the junction of SNARE-JMD or JMD-TMD; (2) charge reversal mutations in STX1A's JMD; and (3) palmitoylation deficiency mutations in STX1A's TMD. We found that both JMD elongations and charge reversal mutations have position-dependent differential effects on $Ca^{2+}$-evoked and spontaneous neurotransmitter release. Importantly, we show that STX1A's JMD regulates the palmitoylation of STX1A's TMD and loss of STX1A palmitoylation either through charge reversal mutation K260E or by loss of TMD cysteines inhibits spontaneous vesicle fusion. Interestingly, the retinal ribbon specific STX3B has a glutamate in the position corresponding to the K260E mutation in STX1A and mutating it with E259K acts as a molecular on-switch. Furthermore, palmitoylation of post-synaptic STX3A can be induced by the exchange of its JMD with STX1A's JMD together with the incorporation of two cysteines into its TMD. Forced palmitoylation of STX3A dramatically enhances spontaneous vesicle fusion suggesting that STX1A regulates spontaneous release through two distinct mechanisms: one through the C-terminal half of its SNARE domain and the other through the palmitoylation of its TMD.

## Editor's evaluation

Exocytosis of synaptic vesicles is mediated by synaptic SNARE proteins that overcome the energy barrier for membrane fusion by assembling into a helical bundle, thus pulling the membranes together. Here the authors have used primary cultures of hippocampal neurons obtained from animals in which the isoforms of syntaxin 1, one of the neuronal SNAREs, are deleted, allowing for the introduction of syntaxin 1a mutants in a clean genetic background. Specifically, the authors investigated mutations into the membrane-proximal region and transmembrane domain of syntaxin 1a and they show that not only charge reversal but also mutations preventing palmitoylation of the transmembrane domain have a strong influence on both spontaneous and evoked neurotransmitter release. The results add important details to our mechanistic understanding of the late steps in SNARE-mediated exocytosis.

## Introduction

Numerous intracellular trafficking pathways utilize various types of vesicle fusion for that the SNARE proteins play a pivotal role. Similarly, synaptic transmission as a means of neuronal communication employs the fusion of the neurotransmitter containing synaptic vesicles (SVs) with the presynaptic plasma membrane. Therefore, presynaptic vesicular release largely relies on the SNARE complex, in this case formed by the presynaptic neuronal SNAREs synaptobrevin-2 (SYB2), SNAP-25 and syntaxin-1A or syntaxin-1B (STX1), assisted by modulatory synaptic proteins (*Rizo, 2018*).

Both STX1 and SYB2 are integral proteins in the plasma and vesicular membranes, respectively, and their C-terminal transmembrane domain (TMD) and SNARE motif are separated only by a short polybasic juxtamembrane domain (JMD). By contrast, SNAP-25 is anchored to the plasma membrane through the palmitoylated cysteines in its linker region between its two SNARE motifs. The N-to-C helical formation of the trans-SNARE complex leads to the apposition of the two membranes (*Gao et al., 2012*; *Sorensen et al., 2006*; *Stein et al., 2009*) in preparation for membrane merger. Additionally, the formation of the cis-SNARE complex on the plasma membrane after vesicle fusion suggests that the SNAREs further zipper into the JMD and TMDs of STX1 and SYB2 (*Hernandez et al., 2012*; *Risselada et al., 2011*; *Stein et al., 2009*).

Importantly, the merger of two lipid membranes is an energetically high-cost process (*Risselada and Mayer, 2020*; *Zhang, 2017*) where the reduction of the energy barrier for membrane merger has been primarily assigned to the modulatory proteins, such as synaptotagmin-1 (SYT1) (*Hui et al., 2009*; *Martens et al., 2007*). Whereas the role of the SNARE domain zippering is well characterized in vesicle fusion, the roles of TMD and JMDs of STX1 and SYB2 are poorly understood as it is still under dispute whether they are active or rather superfluous factors in that process (*Han et al., 2017*).

At this point, it is critical to note that the assumption of the TMDs of STX1 and SYB2 as a passive membrane anchors (*Zhou et al., 2013*) is not compatible with their presumed β-branched nature that confers a high TMD flexibility (*Dhara et al., 2016*; *Hastoy et al., 2017*) and is uncommon among integral transmembrane proteins (*Quint et al., 2010*). Remarkably, another feature shared by STX1 and SYB2 but unusual for the majority of transmembrane proteins is that they are palmitoylated in their TMDs (*Kang et al., 2008*; *Prescott et al., 2009*) which is shown as an alternative mechanism for TMD tilting and flexibility in the membrane (*Blaskovic et al., 2013*; *Charollais and Van Der Goot, 2009*). Whichever mechanism underlies the oblique position of the TMDs of STX1 and SYB2 in the membrane, it is known that their tilted nature causes their polybasic JMDs to be immersed in the membrane, potentially neutralizing the repulsive forces between the apposed vesicular and plasma membrane (*Kim et al., 2002*; *Kweon et al., 2002*; *Williams et al., 2009*). Conceivably, therefore, the JMD and TMD of STX1 and SYB2 might actively regulate vesicle fusion by reducing the energy barrier for membrane merger.

Based on this dispute, we addressed whether STX1A plays additional roles in vesicle fusion to facilitate the membrane merger through its JMD and TMD. We created STX1A mutants where the JMD is elongated or modified by charge reversal single point mutations at different positions that lead to altered electrostatic nature. In addition, we constructed palmitoylation deficient STX1A mutants and analyzed the electrophysiological properties of all STX1A mutants in STX1-null hippocampal mouse neurons. First, we found that the tight coupling of STX1A's JMD to its SNARE domain but not to its TMD is fundamental for neurotransmitter release. Second, and most strikingly, we found that the palmitoylation of STX1A's TMD depends on its JMD's cationic amino acids (AAs) and that loss of palmitoylation dramatically impairs spontaneous release while leaving $Ca^{2+}$-evoked release almost intact. Furthermore, we successfully emulated the regulation of palmitoylation of STX1A's TMD by its JMD and its effect on neurotransmitter release by using other syntaxin isoforms, STX3A or retinal ribbon specific STX3B (*Curtis et al., 2010*; *Curtis et al., 2008*). Based on our data, we propose the direct involvement of STX1A's JMD and TMD in vesicle fusion through electrostatic forces and palmitoylation.

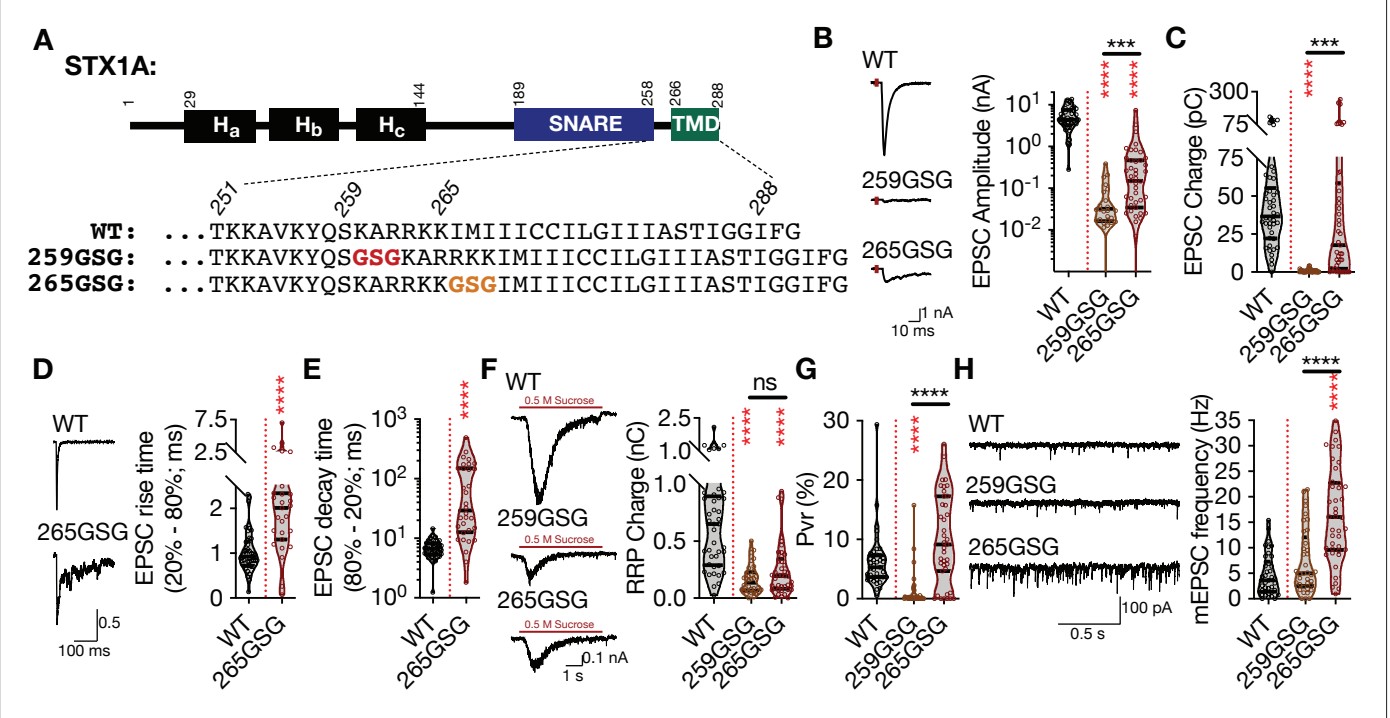

**Figure 1.** The level and mode of impairment of neurotransmitter release by the elongation of STX1A's JMD depends the position of the GSG-insertion. (**A**) STX1A domain structure and insertion of GSG into its JMD. The C-terminal TMD and SNARE motif are separated only by a short polybasic JMD. (**B**) Example traces (left) and quantification of the amplitude (right) of excitatory postsynaptic currents (EPSCs) obtained from hippocampal autaptic STX1-null neurons rescued either with STX1A^WT, STX1A^GSG259, or STX1A^GSG265. (**C**) Quantification of the charge (right) of EPSCs obtained from the same neurons as in (**B**). (**D**) Example traces with the peak normalized to one (left) and quantification of the EPSC rise time measured from 20 to 80% of the EPSC recorded from STX1B^WT or STX1A^GSG265. (**E**) Quantification of the decay time measured from 80 to 20% of the EPSC recorded from the same neurons as in (**D**). (**F**) Example traces (left) and quantification of the charge transfer (right) of 500 mM sucrose-elicited readily releasable pool (RRPs) obtained from the same neurons as in (**B**). (**G**) Quantification of vesicular probability (Pvr) determined as the percentage of the RRP released upon one AP. (**H**) Example traces (left) and quantification of the frequency (right) of mEPSCs recorded at –70 mV. Data information: the artifacts are blanked in example traces in (**B, D,** and **F**). The example traces in (**H**) were filtered at 1 kHz. In (**B–H**), data points represent single observations, the violin bars represent the distribution of the data with lines showing the median and the quartiles. Red and black annotations (stars and ns) on the graphs show the significance comparisons to STX1A^WT and STX1A^GSG259, respectively. Non-parametric Kruskal-Wallis test followed by Dunn's post hoc test was applied to data in (**B, C,** and **F-H**), Mann-Whitney test was applied in (**D** and **E**); ***p≤0.001, ****p≤0.0001. The numerical values are summarized in source data.

The online version of this article includes the following source data for figure 1:

**Source data 1.** Quantification of the neurotransmitter release parameters of STX1-null neurons lentivirally transduced with STX1A^WT or with STX1A JMD elongation mutants.

## Results

### The level and mode of impairment of neurotransmitter release by the elongation of STX1A's JMD depends on the position of the GSG-insertion

The progressive N-to-C zippering of the trans-SNARE complex formed by STX1, SNAP25, and SYB2 sets up the vesicular and plasma membrane in close proximity for vesicle fusion (*Gao et al., 2012*; *Sorensen et al., 2006*; *Stein et al., 2009*). Does the continuity of the SNARE-TMD assembly play a key role in vesicle fusion, and is the distance between the two membranes along the fully zippered trans-SNARE complex important for vesicle fusion? To test this, we followed the classical approach to elongate a helical structure by only one turn, and thus by less than 1 nm. We inserted three AAs, glycine-serine-glycine (GSG), into the JMD of STX1A either at the junction of its SNARE domain and JMD (STX1A^GSG259) or of its JMD and TMD (STX1A^GSG265) (*Figure 1A*), similar to the previous studies (*Hu et al., 2021*; *Kesavan et al., 2007*; *McNew et al., 1999*; *Mostafavi et al., 2017*; *Zhou et al., 2013*). Using our lentiviral expression system in STX1-null neurons (*Vardar et al., 2016*; *Vardar et al., 2021*)

and electrophysiological assessment, we surprisingly found that the insertion of one extra helical turn into the JMD of STX1A led to position-specific physiological phenotypes (*Figure 1*).

First, the amplitude of the excitatory postsynaptic current (EPSC) was reduced to almost zero in both mutants (*Figure 1B*) suggesting that the force transfer from the SNARE complex formation to the merging of the plasma and vesicular membranes is strictly regulated by the length of the JMD of STX1A. On the contrary to the EPSC amplitude, the EPSC charge analysis revealed that STX1A$^{GSG259}$ and STX1A$^{GSG265}$ had differential effects on the vesicle fusion (*Figure 1C*). Whereas, STX1A$^{GSG259}$ completely blocked Ca$^{2+}$-evoked release, STX1A$^{GSG265}$ only slowed it down, as expressed by a twofold and more than tenfold increase in the EPSC rise and the decay time, respectively (*Figure 1D and E*). This suggests that decoupling STX1A's JMD from its SNARE domain has more deleterious effects on Ca$^{2+}$-evoked neurotransmitter release than decoupling it from its TMD. Interestingly, elongation of STX1A's JMD at either position impaired not only Ca$^{2+}$-evoked neurotransmitter release but also the upstream process, namely the vesicle priming, as shown by a significant decrease in the size of the readily releasable pool (RRP) of SVs to ~25% by STX1A$^{GSG259}$ and to ~40% by STX1A$^{GSG265}$ (*Figure 1F*), consistent with previous studies (*Zhou et al., 2013*). This led to an abolishment of vesicular probability (Pvr) in the STX1A$^{GSG259}$ neurons and a trend towards an increase in Pvr in the STX1A$^{GSG265}$ neurons (*Figure 1G*).

Once again, three AA insertions between the SNARE domain and the TMD of STX1A had different effects on spontaneous neurotransmitter release depending on the position of the GSG-insertion. Whereas uncoupling of the SNARE domain and the TMD of STX1A by the GSG259 mutation had no effect on spontaneous neurotransmission, uncoupling of its JMD and the TMD by the GSG265 mutation increased the miniature EPSC (mEPSC) frequency by ~ threefold compared to that of STX1A$^{WT}$ (*Figure 1H*). Importantly, this shows that it is not only the length of the linker region but also the interplay between the SNARE domain-JMD and JMD-TMD that is important for the regulation of synchronous Ca$^{2+}$-evoked and spontaneous release.

## Charge reversal mutations in STX1A's JMD manifest position specific effects on different modes of neurotransmitter release and on molecular weight of STX1A

JMD of STX1A, '260-KARRKK-265', consists of basic residues with characteristics of PIP2 binding motif (*van den Bogaart et al., 2011a*). In fact, it has been shown that it drives PIP2 or PIP3 dependent clustering of STX1A (*Khuong et al., 2013*; *van den Bogaart et al., 2011a*) and the inhibition of its interaction with PIP2/PIP3 leads to defects in neurotransmitter release (*Khuong et al., 2013*). Additionally, the JMD of STX1A is embedded in the membrane due to the tilted conformation of STX1A's TMD (*Kim et al., 2002*; *Kweon et al., 2002*), setting the ground for a possible role in vesicle fusion through electrostatic interactions with the plasma membrane. Therefore, it is plausible that the position-dependent differential effects of the GSG-insertion might be a result of differential perturbations in JMD-membrane interactions. To test this, we introduced single AA charge reversal mutations into the STX1A's JMD (*Figure 2*, *Figure 2—figure supplement 1*). For that purpose, we mutated the lysine or arginine residues from AA 256 to AA 265 into glutamate to achieve maximum alterations in the electrostatic interactions between the JMD and the plasma membrane, where K256E served as a control for the effects of net charge difference only (*Figure 2A*).

Unlike the GSG insertion mutants, charge reversal mutations did not lead to major changes in the EPSC size and kinetics (*Figure 2B, C*, *Figure 2—figure supplement 1*). Overall, STX1A$^{R263E}$ significantly decreased the EPSC amplitude to ~50% compared to that of STX1A$^{WT}$, whereas the other charge reversal mutants only showed a trend towards a 10–35% decrease in EPSC amplitude (*Figure 2B*, *Figure 2—figure supplement 1*). Similarly, not all the charge reversal mutants led to an impairment in vesicle priming but only STX1A$^{K256E}$, STX1A$^{K260E}$ and STX1A$^{R263E}$ mutants reduced the RRP size (*Figure 2D*, *Figure 2—figure supplement 1*). This suggests that the observed impairments in the Ca$^{2+}$-evoked release and vesicle priming are not simply due to the change in the net total charge of STX1A's JMD and that the role of its electrostatic interactions with the plasma membrane is position dependent. Additionally, none of the mutants altered the Pvr (*Figure 2E*, *Figure 2—figure supplement 1*).

Interestingly, charge reversal mutants in STX1A's JMD led to differential results in the spontaneous release. Most of the mutants had no effect on the mEPSC frequency and STX1A$^{R262E}$ showed only a

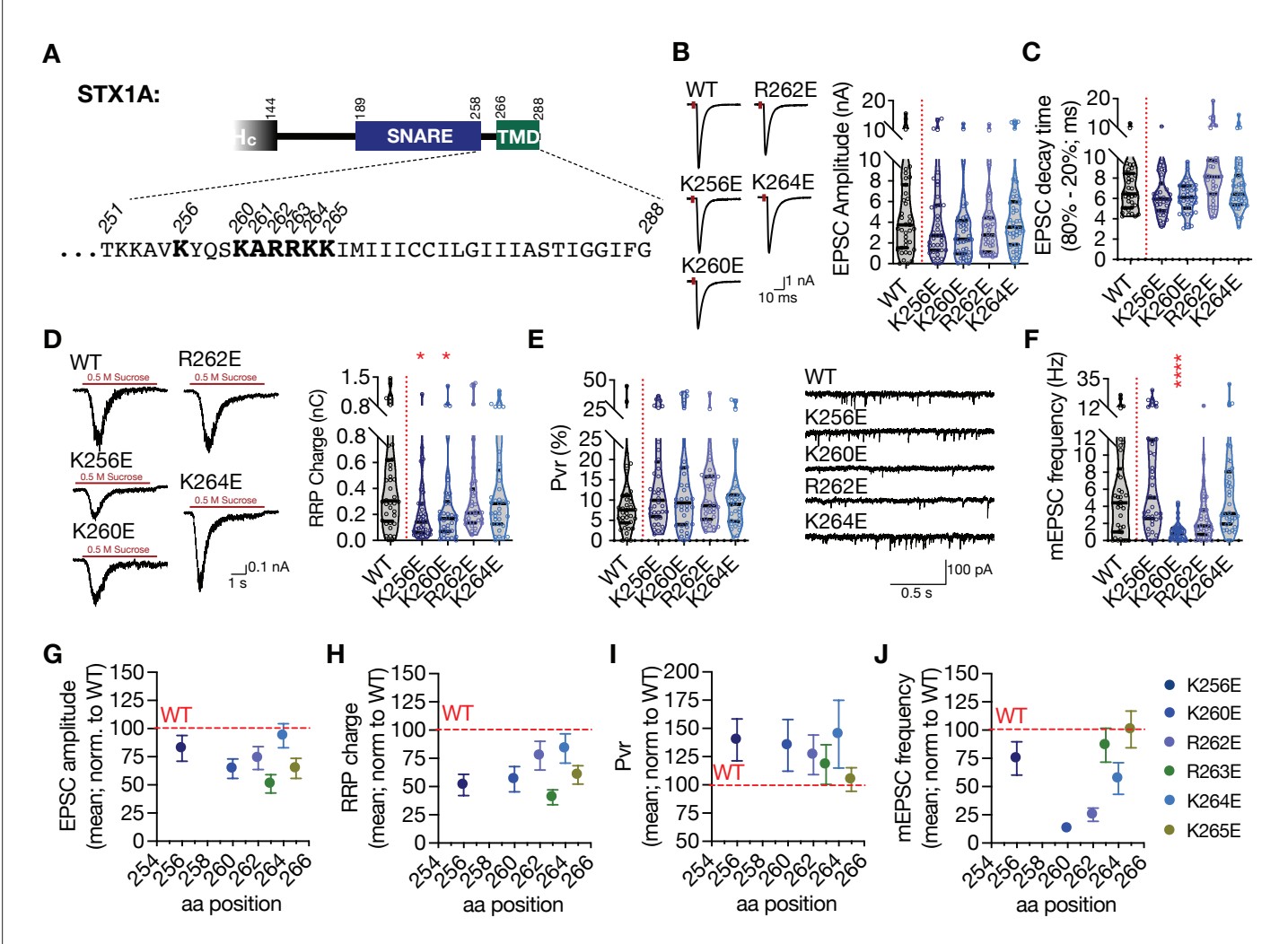

**Figure 2.** Charge reversal mutations in STX1A's JMD manifest position specific effects on different modes of neurotransmitter release. (**A**) Position of the charge reversal mutations on STX1A's JMD. (**B**) Example traces (left) and quantification of the amplitude (right) of excitatory postsynaptic currents (EPSCs) obtained from hippocampal autaptic STX1-null neurons rescued either with STX1A$^{WT}$, STX1A$^{K256E}$, STX1A$^{K260E}$, STX1A$^{R262E}$, or STX1A$^{K264E}$. (**C**) Quantification of the decay time (80–20%) of the EPSC recorded from the same neurons as in (**B**). (**D**) Example traces (left) and quantification of readily releasable pool (RRP) recorded from the same neurons as in (**B**). (**E**) Quantification of vesicular probability (Pvr) recorded from the same neurons as in (**B**). (**F**) Example traces (left) and quantification of the frequency (right) of miniature excitatory postsynaptic currents (mEPSCs) recorded from the same neurons as in (**B**). (**G**) Correlation of the EPSC amplitude normalized to that of STX1A$^{WT}$ to the position of the charge reversal mutation. (**H**) Correlation of the RRP charge normalized to that of STX1A$^{WT}$ to the position of the charge reversal mutation. (**I**) Correlation of Pvr normalized to that of STX1A$^{WT}$ to the position of the charge reversal mutation. Correlation of the mEPSC frequency normalized to that of STX1A$^{WT}$ to the position of the charge reversal mutation. Data information: the artifacts are blanked in example traces in (**B**) and (**D**). The example traces in (**F**) were filtered at 1 kHz. In (**B–F**), data points represent single observations, the violin bars represent the distribution of the data with lines showing the median and the quartiles. In (**G–J**), data points represent mean ± SEM. Red annotations (stars) on the graphs show the significance comparisons to STX1A$^{WT}$. Non-parametric Kruskal-Wallis test followed by Dunn's post hoc test was applied to data in (**B–F**); *p≤0.05, ***p≤0.001, ****p≤0.0001. The numerical values are summarized in source data.

The online version of this article includes the following source data and figure supplement(s) for figure 2:

**Source data 1.** Quantification of the neurotransmitter release parameters of STX1-null neurons lentivirally transduced with STX1A$^{WT}$ or with STX1A JMD charge reversal mutants.

**Figure supplement 1.** Charge reversal mutations in STX1A's JMD manifest position specific effects on different modes of neurotransmitter release.

**Figure supplement 1—source data 1.** Quantification of the neurotransmitter release parameters of STX1-null neurons lentivirally transduced with STX1A$^{WT}$, with STX1A JMD charge reversal mutants, or with STX1A JMD double charge neutralization mutants.

**Figure supplement 2.** Multiple charge neutralization mutations in STX1A's JMD dramatically impairs neurotransmitter release.

trend towards a decrease by ~50% with a p-value of 0.46 compared to that of STX1A$^{WT}$ (*Figure 2F, Figure 2—figure supplement 1*). However, the STX1A$^{K260E}$ mutant unexpectedly showed a dramatic and significant decrease in spontaneous release as it reduced the mEPSC frequency by ~80% from 5.6 Hz to 1 Hz (*Figure 2F*). This again suggests that the perturbation of the electrostatic interactions between the plasma membrane and the JMD of STX1A affects the neurotransmitter release in a position-specific manner. For a better visualization of the position-specific effects of the charge reversal mutations on release parameters, we plotted the EPSC amplitude, RRP size, Pvr, and mEPSC frequency values normalized to the values obtained from STX1A as a function of the AA position of the charge reversal mutations (*Figure 2G-J*). Whereas the alterations in the EPSC amplitude, RRP size, and Pvr showed no correlation to the position of the basic to acidic mutations (*Figure 2G-I*), spontaneous release proved to be specifically perturbed by glutamate insertion at the N-terminus of JMD, that is, STX1A$^{K260E}$ (*Figure 2J*). Importantly, the K256E mutation which resides in the SNARE domain and thus more N-terminally to the JMD showed no impact on the spontaneous release, suggesting a specific function for STX1A's JMD in the regulation of spontaneous neurotransmitter release (*Figure 2F and J*).

It is known that the C-terminal lysine residues K264 and K265 in the JMD of STX1A play a more prominent role in PIP2-binding than the arginine residues R262 and R263 in the middle (*Khuong et al., 2013*). Additionally, STX1A clustering through PIP2/PIP3 has been postulated as an organizing factor of vesicle docking and priming (*Khuong et al., 2013*; *van den Bogaart et al., 2011a*). However, single charge reversal STX1A did not produce a drastic impairment in neurotransmitter release as it would be expected from a mutant unable to mediate vesicle docking and priming. Therefore, it is plausible that the observed phenotypes of single charge reversal mutants might result from a change in the electrostatic nature of the intermembrane area along the SNARE complex and downstream of STX1A/ PIP2 clustering. Based on that, we tested how double or quadruple charge neutralization mutations, STX1A$^{RRAA}$, STX1A$^{KKAA}$, and STX1A$^{4RKA}$ would affect the neurotransmitter release and observed that the inhibition of the putative PIP2 binding site on STX1A indeed reduced the neurotransmitter release to a greater extent compared to that of single charge reversal STX1A mutants (*Figure 2—figure supplement 2*).

Both the SNARE domain and the TMD of STX1A have helical structures, whereas its JMD has an unstructured nature (*Kim et al., 2002*; *Kweon et al., 2002*). This might potentially create a helix-loop-helix formation between the SNARE domain and the TMD when STX1A is isolated. It is known that the mutations potentially affecting the helix-loop-helix formation in the membrane proteins can alter their electrophoresis speed in an SDS-PAGE gel (*Rath et al., 2009*). To test how the charge reversal mutations on STX1A's JMD affect the electrophoretic behavior of STX1A, we probed our mutants on Western Blot (WB) using neuronal lysates obtained from high-density cultures (*Figure 3A*). Surprisingly, STX1A's JMD charge reversal mutations caused not only an apparent different molecular weight on the SDS-PAGE, but they also showed differing band patterns, with the addition of two lower band sizes (*Figure 3A*). To analyze the effect of the STX1A's JMD charge reversal mutations on STX1A's SDS-PAGE behavior, we assigned arbitrary hierarchical numbers from 1 to 6 based on the distance traveled and the number of bands as visualized by STX1A antibody, where number 1 represents the lowest single band as in STX1A$^{K260E}$ and number 6 represents the highest single band as in STX1A$^{WT}$ (*Figure 3B*). We also plotted the weighed band intensity for STX1A for which the lowest band was arbitrarily assigned by 1 × and the highest band was assigned as 100 × as to hierarchically measure the intensity of the bands (*Figure 3C*). We observed a correlation between the SDS-PAGE behavior of STX1A to the position of the charge reversal mutations on its JMD with the K260E mutation causing the most dramatic change in the WB band pattern (*Figure 3B and C*). To test whether the change in the WB band pattern of STX1A lysates is only due to the differential velocity of lysine and glutamate in an SDS-PAGE or whether it reflects the presence or absence of a post-translational modification (PTM), we prepared lysates of HEK293 cell cultures transfected with STX1A$^{WT}$ and charge reversal mutants (*Figure 3D*). Strikingly, none of the constructs showed a comparable pattern to the STX1A obtained from neurons, and the band pattern of all STX1A JMD mutants as well as STX1A$^{WT}$ collapsed to the level of STX1A$^{K260E}$ (*Figure 3D-F*). This suggests that STX1A is post-translationally modified in neurons and this PTM is absent in HEK293 cells. Importantly, it also implies that this PTM is regulated by the basic AA residues in the JMD of STX1A in a position-dependent manner. Does the possible PTM pattern on STX1A correlates with the neurotransmitter release properties? To test this, we plotted

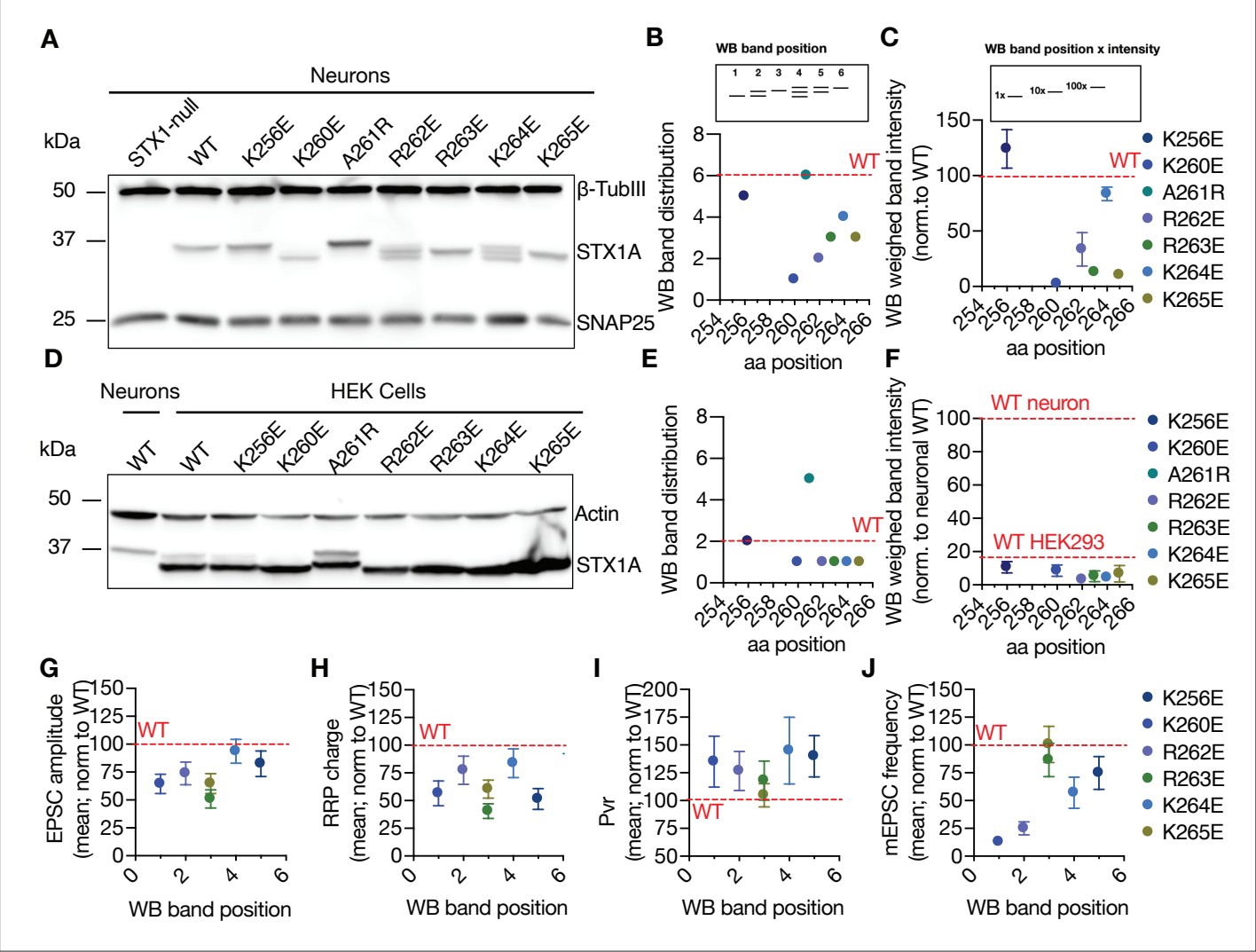

**Figure 3.** Charge reversal mutations in STX1A's JMD manifest position-specific effects on the molecular weight of STX1A. (**A**) Example image of SDS-PAGE of the electrophoretic analysis of lysates obtained from STX1-null neurons transduced with different STX1A JMD charge reversal mutations. (**B**) Quantification of the STX1A band pattern on SDS-PAGE of neuronal lysates through assignment of arbitrary hierarchical numbers from 1 to 6 based on the distance traveled, where number 1 represents the lowest single band as in STX1A$^{K260E}$ and number 6 represents the highest single band as in STX1A$^{WT}$.(**C**) Quantification of the STX1A weighed band intensity of neuronal lysates for which the lowest band was arbitrarily assigned by 1 × and the highest band was assigned as 100 × and multiplied by the measured intensity of the STX1A bands. (**D**) Example image of SDS-PAGE of the electrophoretic analysis of lysates obtained from HEK293 cells transfected with different STX1A JMD charge reversal mutations. (**E**) Quantification of the STX1A band pattern on SDS-PAGE of HEK293 cell lysates as in (**B**). (**F**) Quantification of the STX1A weighed band intensity on SDS-PAGE of HEK293 cell lysates as in (**C**). (**G**) Correlation of the excitatory postsynaptic current (EPSC) amplitude normalized to that of STX1A$^{WT}$ to the western blot (WB) band position of STX1A charge reversal mutation. (**H**) Correlation of the readily releasable pool (RRP) charge normalized to that of STX1A$^{WT}$ to the WB band position of STX1A charge reversal mutation.(**I**) Correlation of vesicular probability (Pvr) normalized to that of STX1A$^{WT}$ to the WB band position of STX1A charge reversal mutation. (**J**) Correlation of the miniature excitatory postsynaptic current (mEPSC) frequency normalized to that of STX1A$^{WT}$ to the WB band position of STX1A charge reversal mutation. Data information: in (**C, F**, and **G–J**), data points represent mean ± SEM. Red lines in all graphs represent the STX1A$^{WT}$ level. The numerical values are summarized in source data.

The online version of this article includes the following source data for figure 3:

**Source data 1.** Quantification of the neurotransmitter release parameters of STX1-null neurons lentivirally transduced with STX1A$^{WT}$ or with STX1A JMD charge reversal mutants –3.

**Source data 2.** Whole SDS-PAGE images represented in *Figure 3A and D*.

the EPSC amplitude, RRP size, Pvr, and the mEPSC frequency values normalized to that of STX1A$^{WT}$ as a function of the WB band distribution of STX1A$^{WT}$ and charge reversal mutants (*Figure 3G-J*). We observed that the lower bands of STX1A in the WB specifically show a correlation with an impairment in spontaneous neurotransmitter release (*Figure 3J*).

## STX1A's JMD modifies the palmitoylation of its TMD which in turn regulates the spontaneous neurotransmitter release

As STX1A charge reversal mutants showed a specific banding pattern on SDS-PAGE when the lysates were obtained from neurons but not from HEK293 cells, we continued with our experiments addressing a potential neuronal specific PTM of STX1A. A modulatory function for a polybasic stretch has been previously shown as a prerequisite for Rac1 palmitoylation (*Navarro-Lérida et al., 2012*). As STX1A is known to be palmitoylated on its two cysteine residues C271 and C272 in its TMD (*Kang et al., 2008*) that neighbors its polybasic JMD, we first tested whether the molecular size shift in STX1A$^{K260E}$ is due to loss of palmitoylation. For that purpose, we created either single point mutants STX1A$^{C271V}$ or STX1A$^{C272V}$ or a double point mutant STX1A$^{CC271,272VV}$ (STX1A$^{CVCV}$) and probed them on SDS-PAGE in comparison to STX1A$^{WT}$ and STX1A$^{K260E}$ (*Figure 4A and B*). Whereas the STX1A$^{CVCV}$ mutant migrated with an electrophoretic speed corresponding to the size of STX1A$^{K260E}$, the single point mutants STX1A$^{C271V}$ and STX1A$^{C272V}$ showed a molecular size in the middle between STX1A$^{WT}$ and STX1A$^{K260E}$ (*Figure 4B*). This suggests that the size shift in the charge reversal mutants (*Figure 3A*) might indeed be due to the impairments in palmitoylation of either cysteine residues (middle bands) or both (lowest band). As a control, we also created a mutant in which CVCV and K260E mutants were combined (STX1A$^{K260E+CVCV}$) and observed that the charge reversal mutation did not cause any further size shift in STX1A$^{CVCV}$ (*Figure 4B*).

To test whether STX1A$^{K260E}$ is palmitoylation deficient, we applied Acyl-Biotin-Exchange (ABE) method in which the palmitate group is exchanged by biotin through hydroxylamine (HAM) mediated cleavage of the thioester bond between a cysteine and a fatty acid chain (*Brigidi and Bamji, 2013*; *Kang et al., 2008*). The lysates of STX1-null neuronal cultures were transduced with STX1A$^{WT}$, STX1A$^{K260E}$, and STX1A$^{CVCV}$ that were N-terminally tagged with FLAG epitope. The lysates of non-transduced STX1-null neuronal cultures were used as a control. During cell lysis, the lysates were additionally treated with N-ethylmaleimide (NEM) solution to block the free thiol groups. STX1 was then pulled down using anti-FLAG magnetic beads. The beads that were attached to STX1 were then incubated in a solution either with or without HAM and subsequently with biotin solution. Now, the covalently bound biotin to the free thiols were exposed after HAM cleavage of the thioester bonds between the palmitate and cysteine and could then be detected by WB using streptavidin antibody. A positive biotin band was detected only in STX1A$^{WT}$ which was treated with the HAM solution (*Figure 4C*, *Figure 4—figure supplement 1*). Neither STX1A$^{K260E}$ nor STX1A$^{CVCV}$ produced any biotin positive bands, showing that both constructs lacked palmitoylation. We further probed the nitrocellulose membranes with STX1A antibody after stripping streptavidin antibody and observed that the lack of detection of biotinylated protein in the groups STX1A$^{K260E}$ and STX1A$^{CVCV}$ was not due to the loss of protein during the ABE-protocol (*Figure 4C*). Superimposition of the images acquired by streptavidin and STX1A antibody treatments showed that the biotin positive band in STX1A$^{WT}$ lysates corresponds to STX1A (*Figure 4—figure supplement 1*).

We then analyzed electrophysiological properties of STX1A$^{C271V}$ and STX1A$^{C272V}$ that seemingly lack one palmitate group as well as STX1A$^{CVCV}$ that lack both of its palmitate groups (*Figure 4B*). EPSCs recorded from STX1A$^{CVCV}$ neurons trended towards a reduction from 6.60 nA to 4.58 nA on average with a p-value of 0.29 compared to that of STX1A$^{WT}$. Neither STX1A$^{C271V}$ nor STX1A$^{C272V}$ showed a similar level of trend towards a reduction as they produced EPSCs of 5.16 nA with p-values of >0.99 and of 0.83, respectively (*Figure 4D*). On the other hand, all the mutants trended towards having faster EPSC kinetics with a faster decay time with p-values ranging from 0.03 to 0.17 compared to that of STX1A$^{WT}$ (*Figure 4E*). Together with an RRP size which is comparable among all the groups (*Figure 4F*), palmitoylation deficient neurons had significantly lower Pvr (*Figure 4G*), suggesting loss of palmitoylation impairs also the efficacy of Ca$^{2+}$-evoked release. Consistent with a reduced Pvr, all palmitoylation deficient mutants showed almost no depression in the short-term plasticity (STP) as determined by 10 Hz stimulation (*Figure 4I*), suggesting an impairment in the vesicular release efficacy. Yet again, the most dramatic effect of palmitoylation deficiency was observed on

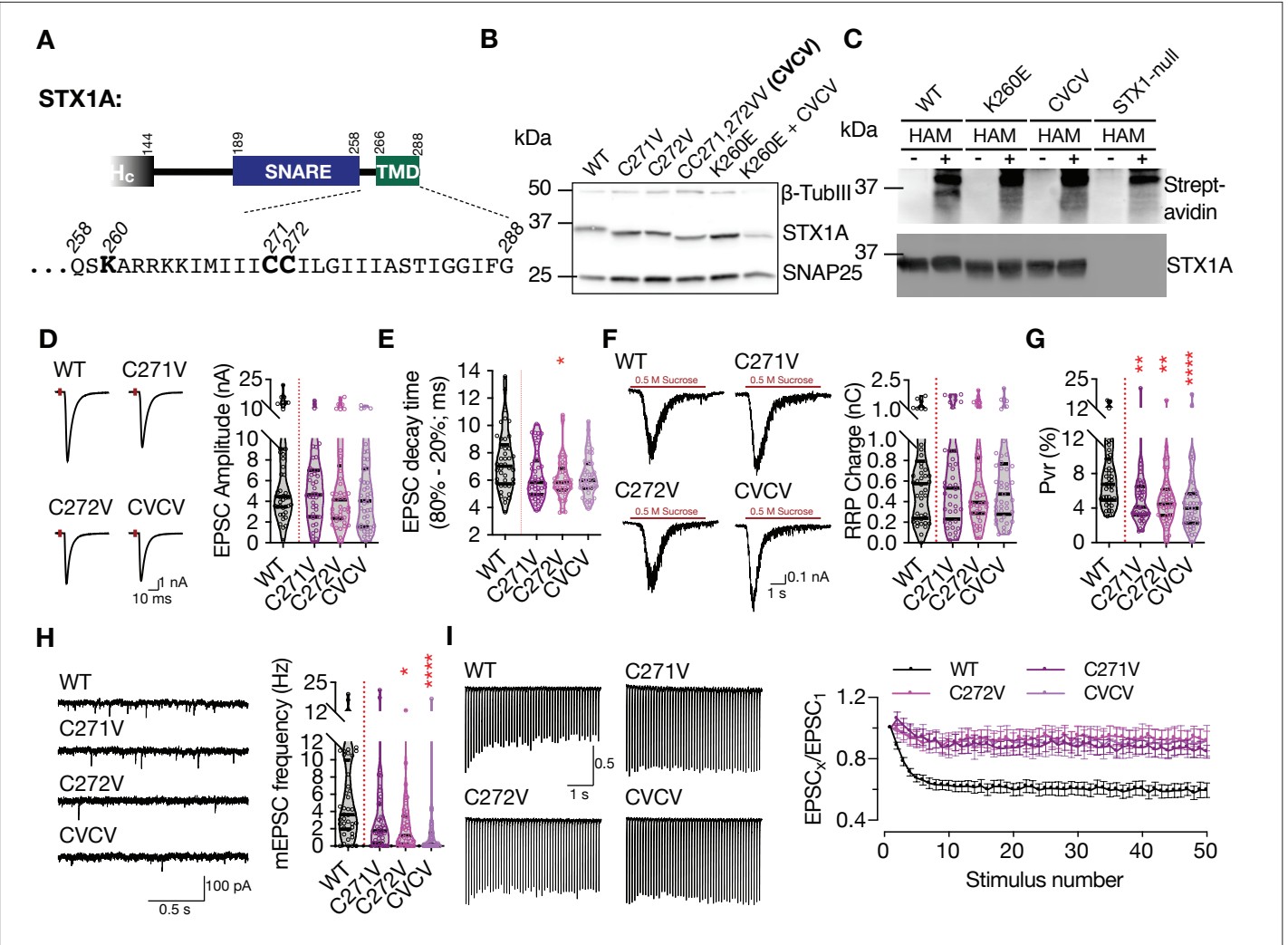

**Figure 4.** Charge reversal mutations in STX1A's JMD manifest position-specific effects on different modes of neurotransmitter release. (A) Position of the palmitoylation deficiency mutations on STX1A's TMD. (B) Example image of SDS-PAGE of the electrophoretic analysis of lysates obtained from STX1-null neurons transduced with STX1A with different palmitoylation deficiency mutations and with STX1A$^{K260E}$ and $^{STX1AK260E+CVCV}$. (C) Example image of the SDS-PAGE of lysates of STX1-null neurons transduced with FLAG-tagged STX1A$^{WT}$, STX1A$^{K260E}$, or STX1A$^{CVCV}$ loaded onto the SDS-PAGE after Acyl-Biotin-Exchange (ABE) method and visualized by Horseradish peroxidase (HRP)-Streptavidin antibody (top panel). After stripping the Streptavidin antibody, the membrane was developed with STX1A antibody (bottom panel). (D) Example traces (left) and quantification of the amplitude (right) of excitatory postsynaptic currents (EPSCs) obtained from hippocampal autaptic STX1-null neurons rescued either with STX1A$^{WT}$, STX1A$^{C271V}$, STX1A$^{C272V}$, or STX1A$^{CVCV}$. (E) Quantification of the decay time (80–20%) of the EPSC recorded from the same neurons as in (D). (F) Example traces (left) and quantification of readily releasable pool (RRP) recorded from the same neurons as in (D). (G) Quantification of vesicular probability (Pvr) recorded from the same neurons as in (D). (H) Example traces (left) and quantification of the frequency (right) of miniature excitatory postsynaptic currents (mEPSCs) recorded from the same neurons as in (B). (I) Example traces (left) and quantification (right) of short-term plasticity (STP) measured by 50 stimulations at 10 Hz recorded from the same neurons as in (B). Data information: the artifacts are blanked in example traces in (D and F). The example traces in (H) were filtered at 1 kHz. In (D–H), data points represent single observations, the violin bars represent the distribution of the data with lines showing the median and the quartiles. In (I), data points represent mean ± SEM. Red annotations (stars) on the graphs show the significance comparisons to STX1A$^{WT}$. Non-parametric Kruskal-Wallis test followed by Dunn's post hoc test was applied to data in (D–H); *p≤0.05, **p≤0.01, ****p≤0.0001. The numerical values are summarized in source data.

The online version of this article includes the following source data and figure supplement(s) for figure 4:

**Source data 1.** Quantification of the neurotransmitter release parameters of STX1-null neurons lentivirally transduced with STX1A$^{WT}$ or with STX1A palmitoylation mutants.

**Source data 2.** Whole SDS-PAGE image represented in *Figure 4B, C*.

**Figure supplement 1.** K260E mutation in STX1A's JMD leads to loss of palmitoylation of its TMD.

**Figure supplement 1—source data 1.** Whole SDS-PAGE images represented in *Figure 4—figure supplement 1*.

spontaneous release as STX1A$^{C271V}$ and STX1A$^{C272V}$ mutants showed either a significant reduction in mEPSC frequency (STX1A$^{C272V}$, p-value of 0.01) or a trend towards it (STX1A$^{C271V}$, p-value of 0.15) (*Figure 4H*). Importantly, the STX1A$^{CVCV}$ mutant which lacks both palmitates showed almost no spontaneous neurotransmitter release (*Figure 4H*), a phenotype similar to the STX1A$^{K260E}$ mutant (*Figure 2F*).

Interestingly, the cysteine residues in STX1A's TMD has been suggested to interact with presynaptic Ca$^{2+}$-channels (*Bachnoff et al., 2013*; *Cohen et al., 2007*; *Sajman et al., 2017*; *Sheng et al., 1994*; *Wiser et al., 1996*) and may inhibit the baseline Ca$^{2+}$-channel activity (*Trus et al., 2001*). This could underly the absence of mEPSC in STX1A$^{CVCV}$ mutant as one mechanism proposed for spontaneous release is the stochastic opening of Ca$^{2+}$-channels in the presynapse (*Kaeser and Regehr, 2014*; *Williams and Smith, 2018*). To test whether loss of palmitoylation by STX1A$^{K260E}$ alone and loss of cysteine residues and their palmitoylation would affect the Ca$^{2+}$-channel activity, we monitored Ca$^{2+}$-influx in the presynapse in the neurons additionally transduced with the Ca$^{2+}$-sensor SynGCampf-6 by stimulating them with different numbers of APs (*Figure 5A-C*). As we have previously reported (*Vardar et al., 2021*), loss of STX1 reduced the global Ca$^{2+}$-influx (*Figure 5A and B*). On the other hand, neither STX1A$^{CVCV}$ nor STX1A$^{K260E}$ showed significantly different SynGCampf-6 signal where the former trended towards an increase for 1AP and the latter trended towards a decrease for 2, 5, 10, and 20 APs (*Figure 5A and C*).

So far, STX1A$^{K260E}$ and STX1A$^{CVCV}$ showed comparable phenotypes in synaptic neurotransmission as they both affected the spontaneous neurotransmitter release more drastically than the Ca$^{2+}$-evoked release (*Figures 2 and 4*). However, only STX1A$^{K260E}$ reduced the size of the RRP but not STX1A$^{CVCV}$ (*Figures 2 and 4*). To uncouple the effects of the synaptic phenotype of K260E by means of the alterations on the membrane proximal electrostatic landscape by charge reversal mutation and by loss of palmitoylation, we created a STX1A construct where the K260E mutation was coupled with CVCV mutation (STX1A$^{K260E+CVCV}$; *Figure 4B*). EPSC amplitudes recorded from STX1A$^{K260E+CVCV}$ neurons were significantly smaller compared to that of STX1A$^{WT}$ but not to that of STX1A$^{K260E}$ (*Figure 5D*). Whereas the RRP size measured from STX1A$^{K260E+CVCV}$ neurons was comparable to that of STX1A$^{K260E}$; Pvr was significantly reduced (*Figure 5F and G*). Most importantly, spontaneous release was again strongly reduced in STX1A$^{K260E+CVCV}$ neurons (*Figure 5H*). Finally, all the STX1A$^{K260E}$, STX1A$^{CVCV}$, and STX1A$^{K260E+CVCV}$ mutants showed impairment in STP as they showed either facilitation or almost no plasticity as opposed to the short-term depression observed in STX1A$^{WT}$ neurons (*Figure 5I*). The observed alteration in the STP was not due to changes in the vesicle fusogenicity as proxied as the fraction of the RRP released by sub-saturating 250 mM sucrose solution (*Figure 5J*).

So far, as STX1A$^{K260E}$ and STX1A$^{CVCV}$ showed only a trend towards a reduction in EPSC amplitude, we pooled all the data obtained from STX1A$^{K260E}$ (*Figures 2, 5 and 6*) and STX1A$^{CVCV}$ (*Figures 4 and 5*) neurons and plotted values normalized to STX1A$^{WT}$ for each individual culture (*Figure 5—figure supplement 1*). The EPSC amplitude was significantly reduced for both STX1A$^{K260E}$ and STX1A$^{CVCV}$ mutants in the pooled data (*Figure 5—figure supplement 1*).

Next, we tested whether the palmitoylation deficiency is due to the loss of the lysine or due to the loss of a basic residue at position AA 260 (*Figure 6A*). For that purpose, we created STX1A mutants in which the lysine K260 on STX1A was exchanged either with a neutral and small alanine (K260A), a neutral glutamine (K260Q) which is more similar to glutamate, or with an arginine (K260R) which is basic. Whereas STX1A$^{K260A}$ produced a banding pattern on SDS-PAGE with two lower bands (*Figure 6B*) similar to STX1A$^{R262E}$ (*Figure 3A*), STX1A$^{K260Q}$ produced three bands (*Figure 6B*) similar to STX1A$^{K264E}$ (*Figure 3A*). On the other hand, STX1A$^{K260R}$ was fully capable of rescuing the banding pattern of STX1A to the highest single band level similar to STX1A$^{WT}$ (*Figure 6B*). Both EPSCs produced by STX1A$^{K260A}$ and STX1A$^{K260Q}$ did not significantly differ from those produced by STX1A$^{K260E}$ (*Figure 6C*). However, STX1A$^{K260R}$ significantly rescued the EPSC back to WT-like level (*Figure 6C*). Similarly, only STX1A$^{K260R}$ neurons had significantly larger RRPs compared to the STX1A$^{K260E}$ neurons (*Figure 6D*) and none of the mutants altered the Pvr (*Figure 6E*). Remarkably, spontaneous release showed a graded improvement in the order of K260E<K260 A<K260 Q<K260 R (*Figure 6F*) comparable to the banding pattern in SDS-PAGE (*Figure 6B*). This suggests that the presence of a basic residue at position 260 in the JMD of STX1A is important for palmitoylation of STX1A in its TMD and therefore the regulation of spontaneous neurotransmitter release (*Figure 6G*).

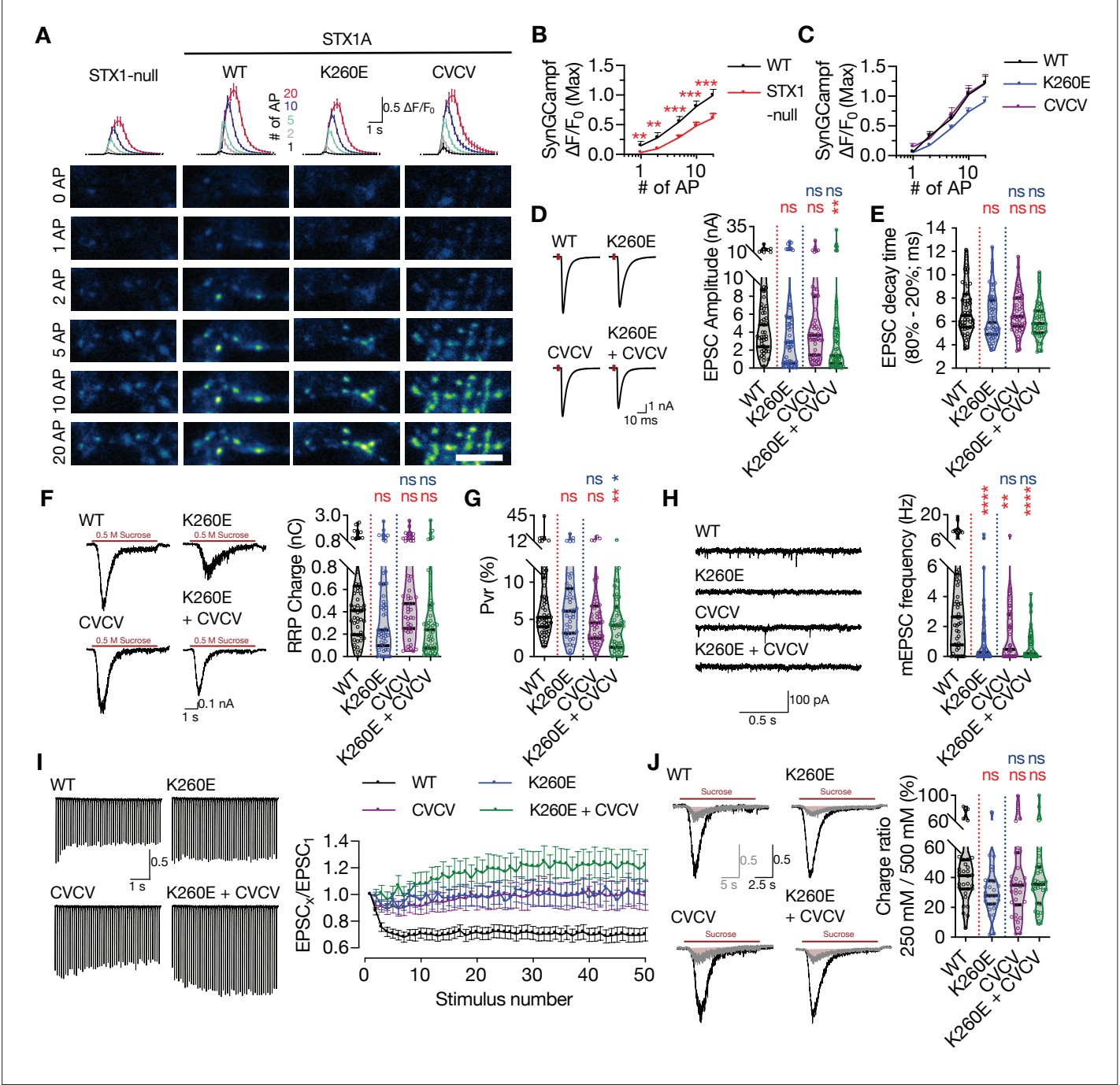

**Figure 5.** Combination of the K260E and CVCV mutations does not further change the phenotype of STX1A[K260E]. (**A**) The average (top panel) of SynGCaMP6f fluorescence as (ΔF/$F_0$) and example images thereof (bottom panels) in STX1-null neurons either not rescued or rescued with STX1A[WT], STX1A[K260E], or STX1A[CVCV]. The images were recorded at baseline, and at 1, 2, 5, 10, and 20 APs. Scale bar: 10 μm. (**B**) Quantification of the SynGCaMP6f fluorescence as (ΔF/$F_0$) in STX1-null neurons either not rescued or rescued with STX1A[WT]. (**C**) Quantification of the SynGCaMP6f fluorescence as (ΔF/$F_0$) in STX1-null neurons rescued with STX1A[WT], STX1A[K260E], or STX1A[CVCV].(**D**) Example traces (left) and quantification of the amplitude (right) of excitatory postsynaptic currents (EPSCs) obtained from hippocampal autaptic STX1-null neurons rescued either with STX1A[WT], STX1A[K260E], STX1A[CVCV], or STX1A[K260E+CVCV]. (**E**) Quantification of the decay time (80–20%) of the EPSC recorded from the same neurons as in (**D**). (**F**) Example traces (left) and quantification of readily releasable pool (RRP) recorded from the same neurons as in (**D**). (**G**) Quantification of vesicular probability (Pvr) recorded from the same neurons as in (**D**). (**H**) Example traces (left) and quantification of the frequency (right) of miniature excitatory postsynaptic currents (mEPSCs) recorded from the same neurons as in (**D**). (**I**) Example traces (left) and quantification (right) of short-term plasticity (STP) measured by 50 stimulations at 10 Hz recorded from same neurons as in (**D**). (**J**) Example traces (left) and quantification (right) of the ratio of the charge transfer triggered by 250 mM

*Figure 5 continued on next page*

*Figure 5 continued*

sucrose over that of 500 mM sucrose recorded from same neurons as in (**D**) as a read-out of fusogenicity of the SVs. Data information: The artifacts are blanked in example traces in (**D, F, and J**). The example traces in (**H**) were filtered at 1 kHz. In (**B, C**, and **I**), data points represent mean ± SEM. In (**D–H** and **J**), data points represent single observations, the violin bars represent the distribution of the data with lines showing the median and the quartiles. Red annotations (stars) on the graphs show the significance comparisons to STX1A^WT. Non-parametric Kruskal-Wallis test followed by Dunn's post hoc test was applied to data in (**B–H and J**); **p≤0.01, ***p≤0.001, ****p≤0.0001. The numerical values are summarized in source data.

The online version of this article includes the following source data and figure supplement(s) for figure 5:

**Source data 1.** Quantification of the neurotransmitter release parameters of STX1-null neurons transduced either with STX1A^WT, STX1A^K260E, STX1A^CVCV, or STX1A^K260E+CVCV.

**Figure supplement 1.** Comparison of the STX1A^K260E and STX1A^CVCV phenotypes by the pooled data of the normalized values for each culture.

**Figure supplement 1—source data 1.** Quantification of the neurotransmitter release parameters of STX1A^K260E and STX1A^CVCV neurons as normalized to the values recorded from STX1A^WT neurons.

## The impacts of palmitoylation of STX1A's TMD on spontaneous release and its regulation by STX1A's JMD can be emulated by using different syntaxin isoforms

So far, we have shown that the basic nature of STX1A's JMD plays an important and differential role in the regulation of the $Ca^{2+}$-evoked and spontaneous release as well as vesicle priming (*Figures 2 and 6*) not only through its electrostatic interactions with the plasma membrane but also through its potential effects on the palmitoylation state of STX1A's TMD. The JMD of STX1s is highly conserved across a number of species yet it shows variability among the different STX isoforms, which are involved in various intracellular trafficking pathways (*Van Komen et al., 2005*).

Among the members of the syntaxin family, only STX1s and STX3B share the same domain structure with a 67% sequence homology and have defined functions in presynaptic neurotransmitter release. Whereas STX1s are expressed in the central synapses that show 'phasic' release, STX3B is the predominant isoform in 'tonically releasing' retinal ribbon synapses where STX1 is excluded (*Curtis et al., 2010*; *Curtis et al., 2008*). Strikingly, STX3B has a glutamate (E) in its JMD at the position 259 leading to the sequence 'EARRKK' (*Figure 7A*). We noted that this is the sequence which renders STX1A incompetent for spontaneous release potentially through its incompatibility for the palmitoylation of STX1A's TMD (*Figures 2, 5 and 6*). Can STX3B carry out neurotransmitter release in a central synapse where STX1s are excluded? If yes, how does a naturally occurring glutamate residue in the most N-terminal residue of its JMD effect the functions of STX3B? To address these questions, we expressed STX3B in STX1-null hippocampal neurons with or without the charge reversal mutation E259K, which produces a STX1A-like JMD in STX3B (*Figure 7A*).

First, we probed STX3B^WT and STX3B^E259K on SDS-PAGE to test whether the E259K mutation changes the banding pattern of STX3B. Indeed, we observed that STX3B^E259K showed a higher molecular weight compared to that of STX3B^WT (*Figure 7B*). This is consistent with the expected palmitoylation deficiency of STX3B^WT due to the presence of a glutamate at the position which corresponds to K260 in STX1A. Before proceeding with our electrophysiological recordings, we also assessed the expression level of STX3B^E259K relative to the expression of STX3B^WT. We determined that both constructs are exogenously expressed at comparable levels at Bassoon-positive puncta when lentivirally introduced into STX1-null neurons (*Figure 7C and D*). Endogenous expression of STX3B in STX1A^WT neurons did not produce a measurable signal at the exposure times of the excitation wavelength used in this study (*Figure 7C and D*).

Next, we tested how the replacement of STX1s either with STX3B^WT or STX3B^E259K affects the neurotransmitter release by measuring the $Ca^{2+}$-evoked and spontaneous release and the RRP of the SVs (*Figure 7E-I*). STX3B^WT was unable to rescue neither the form of neurotransmitter release nor vesicle priming in STX1-null neurons, deeming it dysfunctional in conventional synapses even though it efficiently mediates the neurotransmitter release from retinal ribbon synapses (*Figure 7E-I*). Surprisingly, the E259K mutation served as a molecular on-switch for STX3B as STX3B^E259K fully rescued both $Ca^{2+}$-evoked release with normal release kinetics and spontaneous release (*Figure 7E, F and I*). However, the size of the RRP in STX3B^E259K neurons remained at ~50% of that observed in STX1A^WT neurons (*Figure 7G*), which then led to an increased Pvr (*Figure 7H*). This suggests that the N-terminal lysine of the JMD plays a vital role in the functioning of neuronal syntaxins.

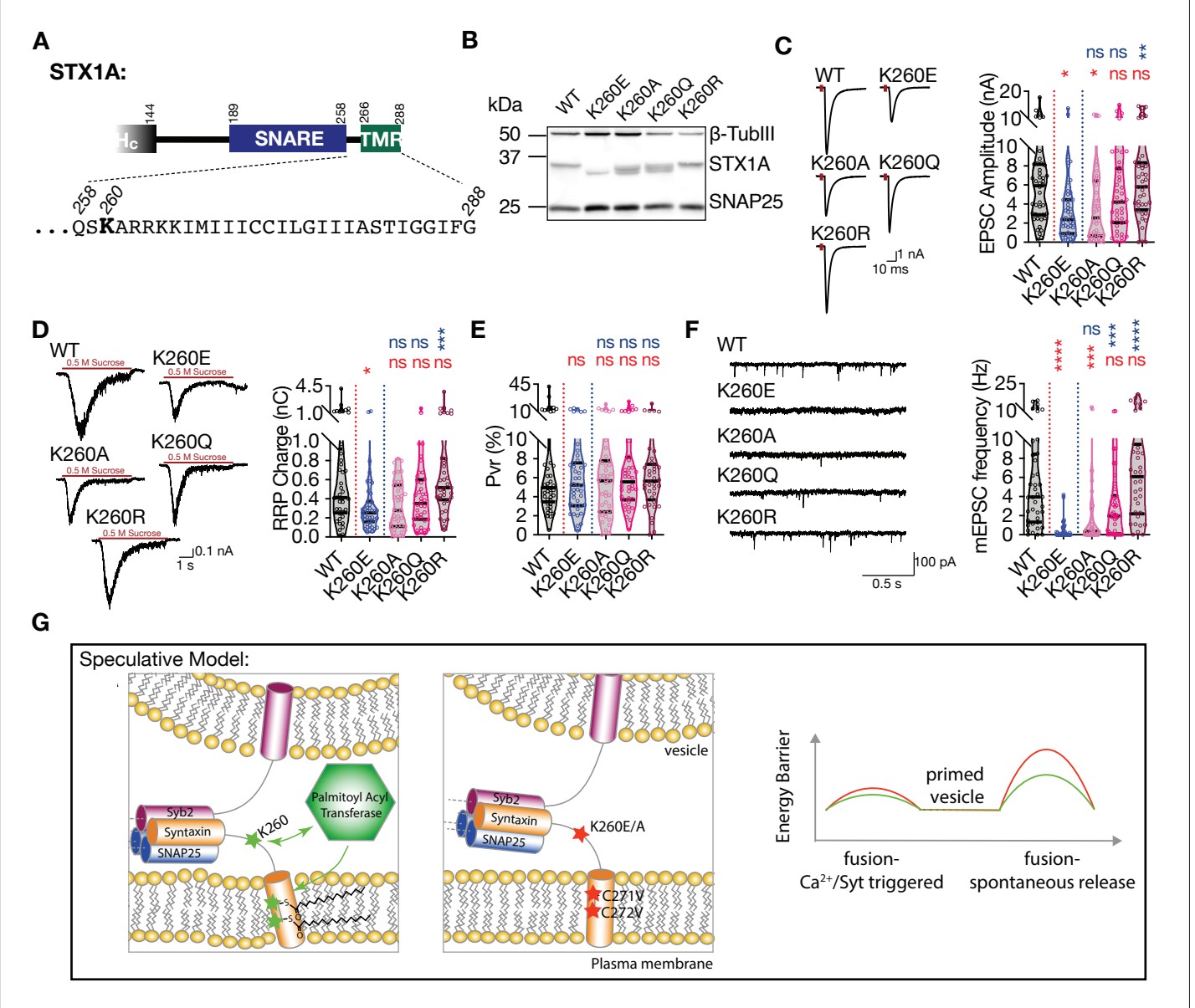

**Figure 6.** Palmitoylation of STX1A's TMD depends on the presence of a basic residue at position AA 260 on its JMD. (**A**) Position of the AA 260 on STX1A's JMD. (**B**) Example image of SDS-PAGE of the electrophoretic analysis of lysates obtained from STX1-null neurons transduced with STX1A^WT, STX1A^K260E, STX1A^K260A, STX1A^K260Q, or STX1A^K260R. (**C**) Example traces (left) and quantification of the amplitude (right) of excitatory postsynaptic currents (EPSCs) obtained from hippocampal autaptic STX1-null neurons rescued either with STX1A^WT, STX1A^K260E, STX1A^K260A, STX1A^K260Q, or STX1A^K260R. (**D**) Example traces (left) and quantification of readily releasable pool (RRP) recorded from the same neurons as in (**C**). (**E**) Quantification of vesicular probability (Pvr) recorded from the same neurons as in (**C**). (**F**) Example traces (left) and quantification of the frequency (right) of miniature excitatory postsynaptic currents (mEPSCs) recorded from the same neurons as in (**C**). (**G**) Speculative model of the role of K260 and C271/C272 residues of STX1A. Left panel: the TMD of STX1A^WT potentially adopts a tilted conformation that reduces the energy barrier for membrane merger. Palmitoylation of the TMD regulated by K260 contributes to its tilted conformation and thus to the facilitation of vesicle fusion. Middle panel: Inhibition of the palmitoylation of STX1A's TMD either by K260E or CVCV mutations encumbers the TMD tilting and thus increases the energy barrier required for membrane merger. Left panel: the energy barrier is lower when STX1A-TMD is palmitoylated (STX1A^WT, green) compared to that when STX1A-TMD is not palmitoylated (STX1A^K260E or STX1A^CVCV, red). Data information: the artifacts are blanked in example traces in (**C and D**). The example traces in (**F**) were filtered at 1 kHz. In (**C–F**), data points represent single observations, the violin bars represent the distribution of the data with lines showing the median and the quartiles. Red and blue annotations (stars and ns) on the graphs show the significance comparisons to STX1A^WT and STX1A^K260E, respectively. Non-parametric Kruskal-Wallis test followed by Dunn's post hoc test was applied to data in (**C–F**); *p≤0.05, **p≤0.01, ***p≤0.001, ****p≤0.0001. The numerical values are summarized in source data.

The online version of this article includes the following source data for figure 6:

*Figure 6 continued on next page*

*Figure 6 continued*

**Source data 1.** Quantification of the neurotransmitter release parameters of STX1-null neurons lentivirally transduced with STX1A[WT] or K260 mutants.

**Source data 2.** Whole SDS-PAGE image represented in *Figure 6B*.

The retinal ribbon synapse specific STX3B is a splice variant of STX3A which is also a neuronal syntaxin with roles indicated in postsynaptic exocytosis (*Jurado et al., 2013*). The differential splicing of STX3A and STX3B occurs in the middle of the SNARE domain generating two products that are identical at their regions between the layer 0 of the SNARE domain and the N-terminus of the protein (*Figure 8—figure supplement 1*). The rest of these proteins spanning the C-terminal half of their SNARE domains, JMD, and TMD show only a 43.1% homology (*Figure 8A*, *Figure 8—figure supplement 1*). Among the sequence differences between the STX3B and STX3A, the JMD and the TMD are of great importance as STX3A lacks the cysteine residues in its TMD and thus the substrate for palmitoylation. Furthermore, its JMD not only has a glutamine at the region corresponding to K260 in STX1A but also has one less basic residue compared to that of STX1A and STX3B (*Figure 8A*, *Figure 8—figure supplement 1*).

Regardless of the SNARE domain, we tested the effects of basic residues in the JMD and the palmitoylation of the TMD on neurotransmitter release. Here we created STX3A mutants in which either two cysteine residues were incorporated into its TMD (STX3A[CC]) or where the 'KARRKK' sequence was introduced into its JMD (STX3A[LINK]) to transmute the region of STX3A into STX1A-like (*Figure 8A*). Furthermore, to evaluate the interplay between the JMD and the palmitoylation of STXs, we also created two additional mutants in which cysteine and JMD incorporation were combined (STX3A[LINK + CC]) or where the whole region spanning the JMD and the TMD of STX3A was exchanged with the corresponding region of STX1A (STX3A[LINK + TMD]) (*Figure 8A*).

We again probed the STX3A[WT] and the STX3A mutants on SDS-PAGE to test for possible alterations in the banding pattern and thereby the palmitoylation state of STX3A (*Figure 7B*). Consistent with our proposal that there is an interplay between the JMD of STX1A and the palmitoylation of its TMD, we observed STX3A[WT] produced one single band and that was not affected when the CC or STX1A's JMD alone were incorporated in STX3A (*Figure 8B and C*). However, a combination of CC and STX1A's JMD in STX3A was enough to reach a partial rescue of the palmitoylation state of STX3A, whereas the introduction of the whole region of STX1A spanning its JMD and TMD effectively reached the full palmitoylation state as manifested by the production of higher molecular weight bands on SDS-PAGE (*Figure 8B and C*). Whereas we could not detect endogenous expression of STX3A in STX1A[WT] neurons at the exposure times used for the excitation wavelength tested, none of the STX3A mutations altered their exogenous expression level at Bassoon-positive puncta compared to that of STX3A[WT] (*Figure 8D and E*).

Remarkably, STX3A[WT], albeit not being a member of any presynaptic neurotransmitter release machinery, supported some level of neurotransmitter release when lentivirally expressed in STX1-null neurons (*Figure 8F*), unlike STX3B[WT] (*Figure 7E*). Whereas the EPSC amplitudes recorded from STX3A[WT] neurons reached ~15% of that of STX1A[WT] neurons (*Figure 8F*), the EPSCs were significantly slowed down, as shown by a doubled duration of both the EPSC rise and the EPSC decay compared to those recorded from STX1A[WT] neurons (*Figure 8G and H*). Introduction of the two cysteine residues or the JMD into STX3A did not lead to any enhancement in the efficacy to support neurotransmitter release, as both the STX3A[CC] and STX3A[LINK] neurons produced EPSCs comparable to STX3A[WT] both in size and kinetics (*Figure 8F-H*). Furthermore, both STX3A[LINK+CC] and STX3A[LINK+TMD] generated EPSCs which were significantly bigger compared to that recorded from STX3A[WT] and which showed only a trend towards a reduction compared to that of produced by STX1A[WT] (*Figure 8F*). Remarkably, the synchronicity of EPSC was rescued partially by STX3A[LINK+CC] and fully by STX3A[LINK+TMD] (*Figure 8G and H*) pointing to the involvement of not only STX1A's TMD in the synchronicity of Ca$^{2+}$-evoked release but also to the JMD and its palmitoylation. Moreover, neurons in which STX1s were exogenously replaced by STX3A[WT] harbored only a very small RRP and that was not rescued by any STX3A mutant (*Figure 8I*) leading to a high Pvr in all neurons expressing STX3A constructs (*Figure 8J*).

Surprisingly, STX3A[WT] neurons spontaneously released SVs at a frequency comparable to that of STX1A[WT] neurons and introduction of the CC or the LINK mutations alone into STX3A showed no alterations in the mEPSC frequency (*Figure 8K*). The combination of the linker region mutation either with CC mutation or with the whole TMD enabled the palmitoylation of the cysteines in the TMD

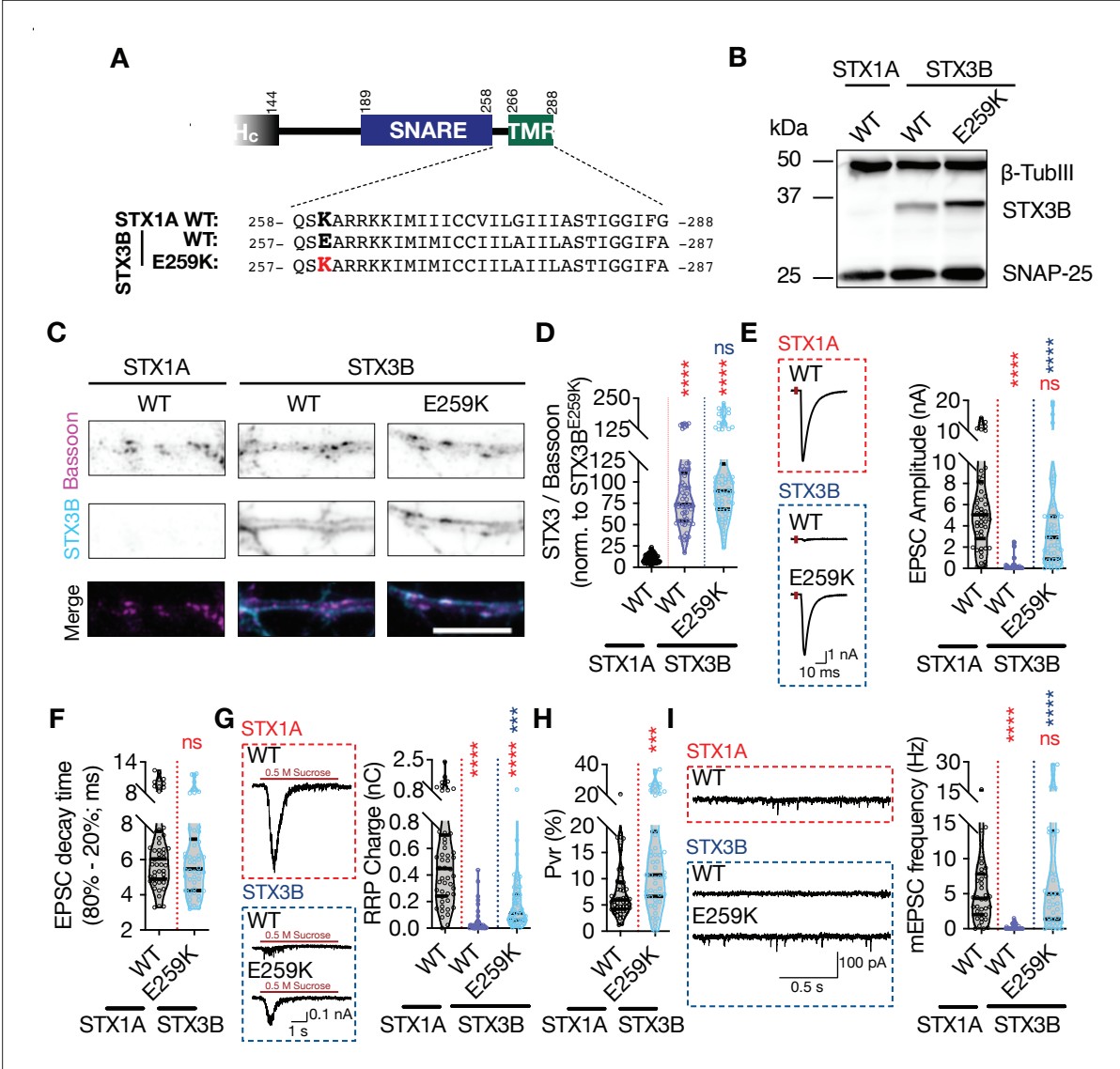

**Figure 7.** Retinal ribbon specific STX3B has a glutamate at position AA 259 rendering its JMD similar to that of STX1AK260E and E259K mutation on STX3B acts as a molecular on-switch. (**A**) Position of the AA 259 on STX3B's JMD. (**B**) Example image of SDS-PAGE of the electrophoretic analysis of lysates obtained from STX1-null neurons transduced with STX1A[WT], STX3B[WT], or STX3B[E259K]. (**C**) Example images of immunofluorescence labeling for Bassoon and STX3B, shown as magenta and cyan, respectively, in the corresponding composite pseudocolored images obtained from high-density cultures of STX1-null hippocampal neurons rescued with STX1A[WT], STX3B[WT], or STX3B[E259K]. Scale bar: 10 μm (**F**) Quantification of the immunofluorescence intensity of STX3B as normalized to the immunofluorescence intensity of Bassoon in the same ROIs as shown in (**C**). The values were then normalized to the values obtained from STX3B[WT] neurons. (**E**) Example traces (left) and quantification of the amplitude (right) of excitatory postsynaptic currents (EPSCs) obtained from hippocampal autaptic STX1-null neurons rescued with STX1A[WT], STX3B[WT], or STX3B[E259K]. (**F**) Quantification of the decay time (80–20%) of the EPSC recorded from the same neurons as in (**E**). (**G**) Example traces (left) and quantification of readily releasable pool (RRP) recorded from the same neurons as in (**E**). (**H**) Quantification of vesicular probability (Pvr) recorded from the same neurons as in (**E**). (**I**) Example traces (left) and quantification of the frequency (right) of miniature excitatory postsynaptic currents (mEPSCs) recorded from the same neurons as in (**E**). Data information: The artifacts are blanked in example traces in (**E and G**). The example traces in (**I**) were filtered at 1 kHz. In (**D–I**), data points represent single observations, the violin bars represent the distribution of the data with lines showing the median and the quartiles. Red and blue annotations (stars and ns) on the graphs show the significance comparisons to STX1A[WT] and STX3B[WT], respectively. Non-parametric Kruskal-Wallis test followed by Dunn's post hoc test was applied to data in (**C–F**); ***p≤0.001, ****p≤0.0001. The numerical values are summarized in source data.

The online version of this article includes the following source data for figure 7:

**Source data 1.** Quantification of the neurotransmitter release parameters of STX1-null neurons lentivirally transduced with STX1A[WT] or with STX3B[WT] or E259K mutant.

**Source data 2.** Whole SDS-PAGE image represented in *Figure 7B*.

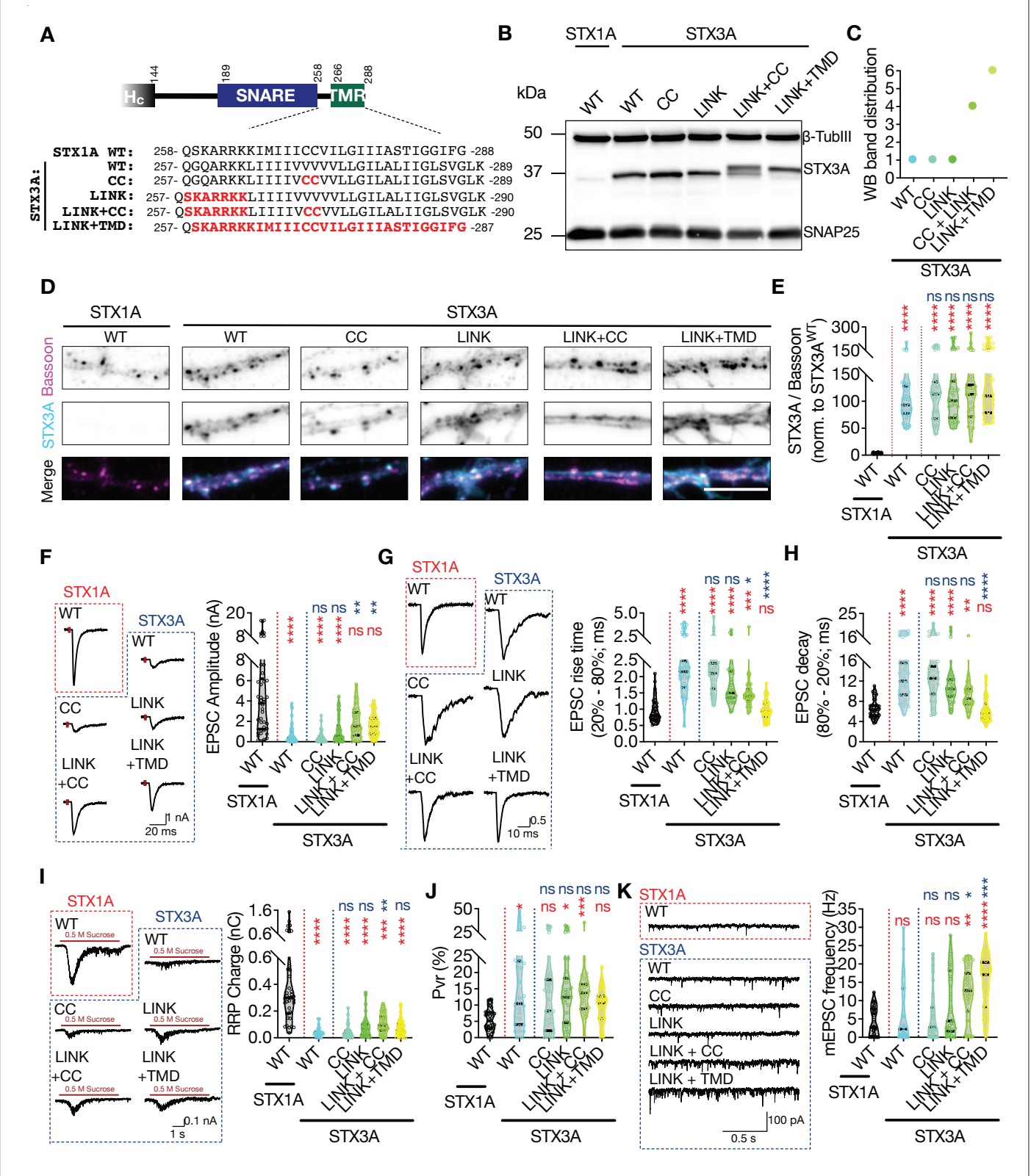

**Figure 8.** The impacts of palmitoylation of STX1A's TMD on spontaneous release and can be emulated by using STX3A. (**A**) Comparison of the JMD and TMD regions of STX1A and STX3A and the mutations introduced into STX3A. (**B**) Example image of SDS-PAGE of the electrophoretic analysis of lysates obtained from STX1-null neurons transduced with STX1A[WT], STX3A[WT], STX3A[CC], STX3A[LINK+CC], or STX3A[LINK+TMR]. (**C**) Quantification of the STX3A band pattern on SDS-PAGE of neuronal lysates through assignment of arbitrary hierarchical numbers from 1 to 6 based on the distance

*Figure 8 continued on next page*

*Figure 8 continued*

traveled as in **Figure 3A**. (**D**) Example images of immunofluorescence labeling for Bassoon and STX3A, shown as magenta and cyan, respectively, in the corresponding composite pseudocolored images obtained from high-density cultures of STX1-null hippocampal neurons rescued with STX1A$^{WT}$, STX3A$^{WT}$, STX3A$^{CC}$, STX3A$^{LINK+CC}$, or STX3A$^{LINK+TMR}$. Scale bar: 10 μm. (**E**) Quantification of the immunofluorescence intensity of STX3A as normalized to the immunofluorescence intensity of Bassoon in the same ROIs as shown in (**D**). The values were then normalized to the values obtained from STX3A$^{WT}$ neurons. (**F**) Example traces (left) and quantification of the amplitude (right) of excitatory postsynaptic currents (EPSCs) obtained from hippocampal autaptic STX1-null neurons rescued with STX1A$^{WT}$, STX3A$^{WT}$, STX3A$^{CC}$, STX3A$^{LINK+CC}$, or STX3A$^{LINK+TMR}$. (**G**) Example traces with the peak normalized to 1 (left) and quantification of the EPSC rise time measured from 20 to 80% of the EPSC recorded from STX1-null neurons as in (**F**). (**H**) Quantification of the decay time (80–20%) of the EPSC recorded from the same neurons as in (**F**). (**I**) Example traces (left) and quantification of readily releasable pool (RRP) recorded from the same neurons as in (**F**). (**J**) Quantification of vesicular probability (Pvr) recorded from the same neurons as in (**F**). (**K**) Example traces (left) and quantification of the frequency (right) of miniature excitatory postsynaptic currents (mEPSCs) recorded from the same neurons as in (**F**). Data information: the artifacts are blanked in example traces in (**F, G**, and **I**). The example traces in (**K**) were filtered at 1 kHz. In (**E–K**), data points represent single observations, the violin bars represent the distribution of the data with lines showing the median and the quartiles. Red and blue annotations (stars and ns) on the graphs show the significance comparisons to STX1A$^{WT}$ and STX3A$^{WT}$, respectively. Non-parametric Kruskal-Wallis test followed by Dunn's post hoc test was applied to data in (**C–F**); *p≤0.05, **p≤0.01, ***p≤0.001, ****p≤0.0001. The numerical values are summarized in source data.

The online version of this article includes the following source data and figure supplement(s) for figure 8:

**Source data 1.** Quantification of the neurotransmitter release parameters of STX1-null neurons lentivirally transduced with STX1A$^{WT}$ or with STX3A$^{WT}$ or mutants.

**Source data 2.** Whole SDS-PAGE images represented in *Figure 8B*.

**Figure supplement 1.** C-terminal half of STX1A's SNARE domain clamps spontaneous neurotransmitter release.

**Figure supplement 1—source data 1.** Quantification of the neurotransmitter release parameters of STX3A and STX3B neurons as normalized to the values recorded from STX1A$^{WT}$ neurons.

that spontaneously discharged the SVs at a frequency of threefold to fourfold of spontaneous release from STX1A$^{WT}$ neurons (**Figure 8K**). This substantial increase in the mEPSC frequency driven by the palmitoylation of the STX3A's TMD essentially points to the same mechanism for the regulation of the spontaneous release by the palmitoylation of STX1A's TMD, as its inhibition either by STX1A$^{K260E}$ or STX1A$^{CVCV}$ diminished spontaneous release (**Figures 2 and 4–6**). This suggests that the prevention of the palmitoylation of the CC residues in the TMD of syntaxins blocks the spontaneous release.

As mentioned above, retinal ribbon synaptic STX3B and postsynaptic STX3A are splice variants which differ only at their C-terminus from the layer 0 of the SNARE domain to the end of the TMD (**Figure 8—figure supplement 1**). The TMD of STX3B is more similar to that of STX1A as the comparative sequence alignment shows that all the AAs at respective positions in the TMD region of STX3B and STX1A share similar or the same biophysical properties (**Figure 8—figure supplement 1**). However, the STX3B$^{E259K}$, which can be considered as the active form of STX3B in conventional synapses, did not show higher spontaneous release efficacy compared to STX1A, even though the cysteine residues are present in that mutant (**Figure 7**). On the other hand, STX3A$^{LINK+TMD}$, which has STX1A's TMD that is now more similar to STX3B, shows an unclamped spontaneous release of SVs (**Figure 8H**). Taking these into account, we compared the AA sequences of STX1A, STX3B$^{E259K}$, and STX3A$^{LINK+TMD}$ and plotted the mEPSC frequencies as values normalized to the STX1A$^{WT}$ for each individual culture (**Figure 8—figure supplement 1**). The sequence alignment shows that the STX3A$^{LINK+TMD}$ mutant which spontaneously releases SVs at the highest frequency differs from both STX1A$^{WT}$ and STX3B$^{E259K}$ only in the C-terminal half of its SNARE domain (**Figure 8—figure supplement 1**). This suggests that the C-terminal half of STX1A's SNARE domain from its layers 0–8 plays a pivotal role in the clamping of the spontaneous vesicle fusion.

A hypothesis in account of spontaneous vesicle fusion suggests that the vesicles close to the Ca$^{2+}$-channels fuse with the membrane upon stochastic opening of the Ca$^{2+}$-channels (**Kaeser and Regehr, 2014**; **Williams and Smith, 2018**). Therefore, alterations in the regulation of Ca$^{2+}$-channel gating, such as proposed inhibition of baseline activity of Ca$^{2+}$-channels by the cysteine residues of STX1A's TMD (**Trus et al., 2001**), could lead to a decrease or increase in the spontaneous neurotransmitter release as a result of altered spontaneous Ca$^{2+}$-influx into the synapse. Desynchronization of the Ca$^{2+}$-evoked neurotransmitter release by STX3A$^{WT}$ might be a result of uncoupling of the Ca$^{2+}$-channel-SV spatial organization. In that scenario, it is plausible that the TMD of STX1A plays a role in the Ca$^{2+}$-channel-SV coupling, as its incorporation into STX3A leads to the recovery of synchronous release (**Figure 8G and H**). To test whether and how STX3B and STX3A affect the global Ca$^{2+}$-influx in their WT or mutant

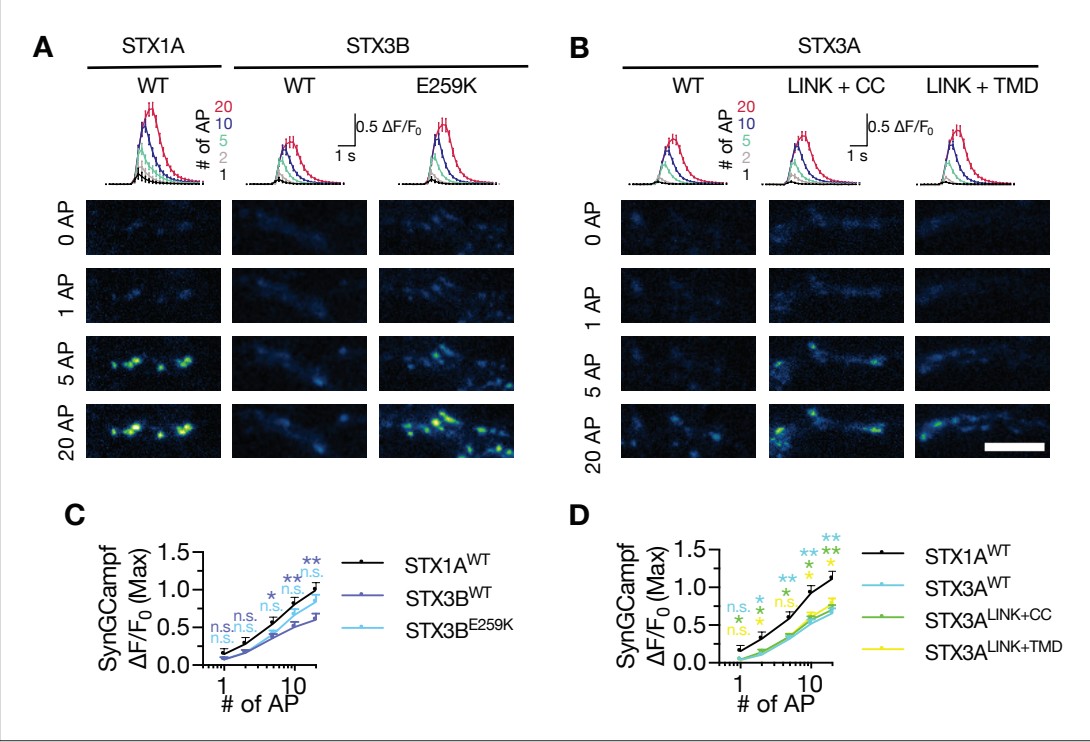

**Figure 9.** Neither STX3A nor STX3B rescues the global Ca2+-influx back at WT-like level in STX1-null neurons. (**A**) The average (top panel) of SynGCaMP6f fluorescence as ($\Delta F/F_0$) and example images thereof (bottom panels) in STX1-null neurons rescued with STX1A[WT], STX3B[WT], or STX3B[E259K]. The images were recorded at baseline, and at 1, 2, 5, 10, and 20 APs. Scale bar: 10 µm. (**B**) The average (top panel) of SynGCaMP6f fluorescence as ($\Delta F/F_0$) and example images thereof (bottom panels) in STX1-null neurons rescued with STX1A[WT], STX3A[WT], STX3A[CC], STX3A[LINK+CC], or STX3A[LINK+TMR]. The images were recorded at baseline, and at 1, 2, 5, 10, and 20 APs. Scale bar: 10 µm. (**C**) Quantification of the SynGCaMP6f fluorescence as ($\Delta F/F_0$) in STX1-null neurons rescued with STX1A[WT], STX3B[WT], or STX3B[E259K]. (**D**) Quantification of the SynGCaMP6f fluorescence as ($\Delta F/F_0$) in STX1-null neurons rescued with STX1A[WT], STX3A[WT], STX3A[CC], STX3A[LINK+CC], or STX3A[LINK+TMR]. Data information: in (**C and D**), data points represent mean ± SEM. All annotations (stars and ns) on the graphs show the significance comparisons to STX1A[WT] with the color of corresponding group. Non-parametric Kruskal-Wallis test followed by Dunn's post hoc test was applied to data in (**C and D**); *p≤0.05, **p≤0.01. The numerical values are summarized in source data.

The online version of this article includes the following source data for figure 9:

**Source data 1.** Quantification of the presynaptic Ca$^{2+}$-influx in STX1-null neurons transduced either with STX1A[WT], STX3B[WT] or mutant, or STX3A[WT] or mutants.

forms, we again co-expressed the Ca$^{2+}$-sensor SynGCampf6 in the autaptic synapses and measured the Ca$^{2+}$-influx upon differing numbers of APs (**Figure 9**). Interestingly, both STX3B[WT] and STX3A[WT] failed to rescue the reduction of the Ca$^{2+}$-influx in the presynaptic terminals due to STX1-loss (**Figure 9A and B**). However, STX3B[E259K] rescued the global Ca$^{2+}$-influx when higher numbers of APs were elicited, suggesting that syntaxins might play a role in the gating of Ca$^{2+}$-channels (**Figure 9A**). STX3A mutants did not have any impact on Ca$^{2+}$-influx as it remained at STX1-null levels even for the mutants that rescued the EPSC synchronicity and led to an excessive amount of spontaneous neurotransmitter release (**Figure 9B**). As the Ca$^{2+}$-influx in neurons expressing STX3A variants remained low at any AP number compared to that of STX1A neurons, it is likely that the overall Ca$^{2+}$-trafficking and hence Ca$^{2+}$-abundance might be impaired in the absence of STX1 which cannot be rescued by STX3A.

## Discussion

The presynaptic SNARE complex formation by STX1A, SYB2, and SNAP-25 set up the vesicular and plasma membrane in close proximity through the N-to-C zippering of their SNARE domains (**Gao et al., 2012**; **Sorensen et al., 2006**; **Stein et al., 2009**). In our study, we addressed whether STX1A plays an additional role in vesicle fusion to facilitate the membrane merger through its JMD and TMD. Based on our data we propose that the JMD of STX1A regulates the membrane merger through adjusting the distance and the electrostatic nature of the inter-membrane area along the trans-SNARE

complex. Whereas STX1A's JMD also determines the palmitoylation state of STX1A's TMD, the palmitoylation of STX1A's TMD might directly influence the energy barrier for membrane merger, which is specifically apparent for spontaneous release. We conclude that the JMD and TMD of STX1A function not only as a membrane anchor but are actively involved in the regulation of vesicle fusion.

## The role of STX1A's JMD in neurotransmitter release extends beyond its interaction with PIP2

The current models of SV fusion have given a significant role to the JMD of STX1A, where it drives STX1A clustering through its interaction with PIP2/PIP3 (*Khuong et al., 2013*; *van den Bogaart et al., 2011a*) and therefore indirectly regulates the spatial organization of vesicle docking as PIP2 also binds to SYT1 (*Aoyagi et al., 2005*; *Chen et al., 2021*; *Honigmann et al., 2013*; *Park et al., 2015*). Interestingly, the specific basic residues K256, K260, and R263 that are 3 or 4 AAs apart have the greatest impact on the formation of the pool of releasable vesicles (*Figure 2*). As these residues potentially reside on the same side of an alpha-helical structure formed by the SNARE zippering, it can be argued that the priming of synaptic vesicles might be regulated through their electrostatic interactions with the negatively charged head groups of phospholipids. Furthermore, it is important to note that the two C-terminal lysine residues in STX1A's JMD play a key role in PIP2 binding (*Murray and Tamm, 2011*) as neutralization of both shows a more dramatic impairment not only in vesicle fusion but also in vesicle priming in mouse (*Figure 2—figure supplement 2*) and fly neurons (*Khuong et al., 2013*) compared to the charge reversal mutations of either K264 or K265 (*Figure 2*). This suggests that the general electrostatic nature of the juxtamembrane area determined by the JMD of STX1A, as well as by SYB2's JMD (*Williams et al., 2009*) directly influences the neurotransmitter release in a process downstream of STX1A-PIP2 binding.

Our data also show that the function of STX1A's JMD is not limited to the regulation of the electrostatic nature of the intermembrane area, but also possibly to the coordination of the intermembrane distance along the trans-SNARE complex. The slowing down of the release together with the unclamping of spontaneous release by STX1A$^{GSG265}$ (*Figure 1*) is reminiscent of the phenotype of the loss of the Ca$^{2+}$-sensor (*Chang et al., 2017*; *Courtney et al., 2019*; *Vevea and Chapman, 2020*; *Xu et al., 2009*). Through its interactions with the lipids, SYT1 primarily assists the closure of the gap between the two membranes for tight docking of the SVs (*Araç et al., 2006*; *Chang et al., 2017*; *Chen et al., 2021*; *van den Bogaart et al., 2011b*) and thereby accelerates the Ca$^{2+}$-evoked fast synchronous release (*Geppert et al., 1994*; *Littleton et al., 1993*). Thus, it is possible that an increase in the intermembrane distance along the trans-SNARE complex by an elongation of the STX1A's JMD, even by one helical turn that is less than 1 nm, might impair SYT1's function as a distance regulator in synchronizing the vesicular release.

It is not clear, however, why STX1A$^{GSG259}$ has a more deleterious effect on Ca$^{2+}$-evoked release than STX1A$^{GSG265}$ does, but it is conceivable that it blocks the helical continuity of the SNARE complex into the JMD of STX1A and SYB2. This phenomenon of asymmetrical impact on the SNARE-JMD or JMD-TMD decoupling of vesicle fusion is not unique to STX1A as it is also observed in SYB2 mutants where the JMD is also elongated at different positions (*Hu et al., 2021*; *Kesavan et al., 2007*; *McNew et al., 1999*; *Mostafavi et al., 2017*). Likewise, the helix breaking proline insertion in SYB2 at the junction of its SNARE-JMD, but not at the junction of its JMD-TMD, is detrimental to liposome fusion (*Hu et al., 2021*) suggesting an essential mechanistic similarity between the regulation of the membrane merger by the JMD of the plasma and the vesicular SNAREs. Given that the spontaneous release was either unaffected or facilitated by the elongation of STX1A's JMD (*Figure 1*), it appears that the JMD of both STX1A and SYB2 may contribute to the vesicle fusion by regulating the electrostatic nature and the distance of the intermembrane area along the trans-SNARE complex and the helical continuity of the SNARE complex into the SNARE-JMD in a Ca$^{2+}$-dependent manner.

## Palmitoylation of STX1A's TMD potentially reduces the energy barrier required for membrane merger

Palmitoylation as a PTM is used for various functions for a wide panel of presynaptic and postsynaptic proteins (*Kang et al., 2008*; *Matt et al., 2019*; *Naumenko and Ponimaskin, 2018*; *Prescott et al., 2009*). First, it anchors soluble proteins, such as SNAP-25 and CSPα, onto their target membranes (*Prescott et al., 2009*), which is generally considered the primary function of palmitoylation. However,

palmitoylation of the integral membrane proteins such as STX1A, SYB2, and SYT1, has less defined functions (*Prescott et al., 2009*). It is known that palmitoylation dependent alteration of a TMD's length can affect its hydrophobic mismatch status with the corresponding carrier membrane (*Greaves and Chamberlain, 2007*) and therefore it can be utilized for the hydrophobic mismatch regulated transport of an integral membrane protein through Golgi-complex (*Ernst et al., 2018*; *Ernst et al., 2019*). Based on the same mechanism, palmitoylation can also determine the final localization and the clustering of an integral membrane protein, particularly in cholesterol-rich, thick, and rigid membranes that do not conform to the general properties of TMDs (*Levental et al., 2010*; *Melkonian et al., 1999*; *van Duyl et al., 2002*). In fact, STX1A forms clusters in the membrane that depend on both its cholesterol (Lang et al.; *Murray and Tamm, 2009*) and PIP2 content (*Honigmann et al., 2013*; *Murray and Tamm, 2009*). Furthermore, SYB2, which is also palmitoylated in its TMD (*Kang et al., 2004*; *Veit et al., 2000*), is associated with cholesterol-rich lipid rafts derived from SVs (*Chamberlain et al., 2001*; *Chamberlain and Gould, 2002*), which affects its SNARE domain and TMD conformation (*Han et al., 2016*; *Tong et al., 2009*; *Wang et al., 2020*).

Taken together with the reported reduction of $Ca^{2+}$-evoked neurotransmitter release upon cholesterol depletion (*Lang et al., 2001*; Wasser et al.), it is tempting to speculate that the impacts of palmitoylation deficiency of STX1A's TMD might stem from a general localization, oligomerization and mobility defect of STX1A, similar to the cause of neurotransmitter release deficiency in neurons expressing non-palmitoylated SYT1 (*Kang et al., 2004*). However, such a localization and mobility defect of STX1A would be detrimental not only to spontaneous neurotransmitter release, but to all types of vesicle fusion and vesicle priming. Yet, our palmitoylation deficient STX1A mutants do not manifest a drastic change in vesicle priming and $Ca^{2+}$-evoked vesicle fusion (*Figure 4*) and chemical removal of cholesterol has been shown to increase spontaneous vesicle fusion (*Wasser et al., 2007*). This renders the hypothesis of mislocalized STX1A due to the loss of palmitoylation unlikely. As loss of complete or individual palmitoylation sites on SNAP25 does not compromise its proper localization (*Greaves et al., 2009*; *Vogel et al., 2000*; *Washbourne et al., 2001*) but vesicle exocytosis (*Washbourne et al., 2001*), it can be argued that the palmitoylation of the trans-SNARE proteins has a direct influence on the membrane merger.

A membrane merger is an energetically costly process, and in central synapses, several mechanisms are employed to lower the energy barrier for the SV fusion. These mechanisms include—besides the SNARE complex formation—the concerted actions of SYT1 and complexin (CPX), which confer the central synapse with a spatial and temporal acuity of SV release upon an incoming $Ca^{2+}$-signal (*Brunger et al., 2018*; *Rizo, 2018*). Furthermore, the TMD of SYB2 itself can facilitate the membrane merger through its tilted conformation with a high flexibility in the membrane that largely stems from its distinctively high propensity for β-branches (*Dhara et al., 2016*; *Hastoy et al., 2017*). Whether or not STX1A's TMD, too, shows a higher propensity for β-sheets than for α-helices is not known, but is likely, as it is rich in β-branching isoleucine (*Figure 1*). Remarkably, a similar function in TMD tilting has been attributed to the palmitoylation of membrane proteins (*Blaskovic et al., 2013*; *Charollais and Van Der Goot, 2009*). In that light, we propose that the palmitoylation of STX1A's TMD serves as another mechanism for the lowering of the energy barrier for membrane fusion. This mechanism might seem particularly important for spontaneous vesicle fusion at first glance. However, the reduction of the $Ca^{2+}$-evoked release albeit being small, emphasizes a general impairment in the vesicle fusion due to loss of palmitoylation. This is more evident in the decrease of the vesicular release probability especially revealed by the impairment in STP elicited by a 10 Hz stimulation in neurons that express STX1A single or double palmitoylation mutants as well as STX1A$^{K260E}$ (*Figures 4 and 5*).

Furthermore, support for our hypothesis that palmitoylation of STX1A's TMD reduces the energy barrier for membrane merger (*Figure 6*, speculative model) comes from our observation that the incorporation of the palmitoylation substrate cysteines into the TMD of STX3A exclusively facilitates spontaneous fusion even when the rest of STX3A's TMD is kept unchanged (*Figure 8*). According to one line of evidence, spontaneous and $Ca^{2+}$-evoked vesicle fusion share the same mechanism and membrane topology and thus the stochastic opening of $Ca^{2+}$-channels enable the spontaneous fusion of single SVs that are in a 'fusion-ready' state (*Kaeser and Regehr, 2014*; *Williams and Smith, 2018*). As STX1A has been postulated to inhibit baseline $Ca^{2+}$-channel activity through its cysteine residues in its TMD (*Bachnoff et al., 2013*; *Cohen et al., 2007*; *Trus et al., 2001*), it is possible that the reduction of the spontaneous neurotransmitter release might be due to the inhibition of the

stochastic opening of Ca²⁺-channels. We cannot rule out this scenario based on our data. However, our AP-evoked presynaptic global Ca²⁺-influx assay shows that the STX3A is unable to rescue the Ca²⁺-channel abundance and/or activity when expressed in STX1-null neurons in either WT or any mutant form and the Ca²⁺-influx remains smaller than in STX1A-WT neurons even at a high number of AP elicited (*Figure 9*). Yet, mutant STX3A that has two cysteine residues in its TMD combined with STX1A's JMD or that has the entire JMD and TMD of STX1A leads to threefold to fourfold increase in spontaneous release. This points to an additional role of the STX1A's JMD-TMD in the facilitation of vesicle fusion besides its putative interaction with Ca²⁺-channels (*Bachnoff et al., 2013*; *Cohen et al., 2007*; *Trus et al., 2001*). Taking this into account, we propose that natural palmitoylation of STX1A's TMD and forced palmitoylation of STX3A's TMD likely increase the number of SVs that more easily overcome the energy barrier for membrane merger (*Figure 6*, speculative model).

## STX1A's TMD is required for the synchronicity of the Ca²⁺-evoked release

Our studies using the charge reversal mutations on the JMD and the palmitoylation mutations on the TMD of STX1A show that these mutants can reach the AZ, as all of them could mediate some sort neurotransmitter release (*Figures 2 and 4–6*). However, a potential effect of STX1A's TMD for fine tuning of its localization is plausible, as the differential distribution of other syntaxins, STX3A and STX4, in the plasma membrane is based on their TMD length (*Bulbarelli et al., 2002*; *Watson and Pessin, 2001*). Consistently swapping STX3A's TMD with STX1A's TMD was sufficient to rescue the synchronous release (*Figure 8*) suggesting a role for STX1A's TMD in proper Ca²⁺-secretion-vesicle fusion coupling. However, the TMD of the retinal ribbon synapse syntaxin, STX3B, is almost identical to STX1A's TMD with only two AA difference in the sequence (*Figure 8—figure supplement 1*); further pointing out the importance of the TMD for syntaxins involved in neurotransmitter release either from conventional or ribbon synapses.

STX3B's JMD differs from that of STX1A as it resembles the mutant STX1A^K260E and the STX3B^E259K mutation, that acts as a molecular on-switch in the STX1-null hippocampal synapses (*Figure 7*). It is not yet clear why the alteration of the STX3B's but not STX1A's JMD has such a dramatic effect also on Ca²⁺-evoked vesicle fusion and vesicle priming. Yet, different syntaxins might go through different conformational changes due to different intramolecular interactions with their regulatory N-terminal domains. Indeed, STX3B's opening in retinal ribbon synapses, where Munc13s do not function as a primary vesicle priming factor (*Cooper et al., 2012*), is mediated through its N-peptide phosphorylation (*Campbell et al., 2020*; *Liu et al., 2014*). Whereas such an opening mechanism has not been reported for STX1A, the JMD has been proposed to induce conformational changes on STX1A through interactions with PIP2 which in turn leads to the phosphorylation of its N-peptide on S14 (*Khelashvili et al., 2012*; *Singer-Lahat et al., 2018*). Therefore, it is conceivable that the E259K mutation's impact on STX3B as a molecular on-off switch might be due to the opening of STX3B through phosphorylation of its N-peptide.

Whether or not STX3B is palmitoylated in the retinal ribbon synapses is unknown but the presence of cysteines in its TMD hints at a potential for palmitoylation. On the other hand, it is also possible that the E259 might render the TMD of STX3B inadequate for palmitoylation as in the case of STX1A^K260E. This raises the possibility of inhibition of spontaneous vesicle fusion in retinal ribbon synapses by the non-palmitoylated TMD of STX3B. This is interesting as the retinal ribbon synapses operate in an essentially different manner compared to the conventional synapses as they predominantly rely on the tonic fusion but not on AP-driven phasic fusion of the SVs. Thus, JMD dependent inhibition of the TMD palmitoylation of STX3B and therefore the blockage of the spontaneous release even at the expense of the full capacity of Ca²⁺-evoked release might be beneficial for a ribbon synapse to enhance the signal-to-noise ratio of the tonic neurotransmitter release.

Our chimeric analysis of the TMD of different syntaxins also shows that besides the regulation of the efficacy of SV fusion by the TMD of STX1A, spontaneous SV fusion is further modulated by the C-terminal half of STX1A's SNARE domain, suggesting a direct involvement of STX1A in the SV clamp. Which intermolecular interactions regulate the SNARE domain mediated clamping of spontaneous release is not clear, as known interactions of STX1A with the modulatory proteins SYT1 and CPX are shown to be carried out through the N-terminal half of its SNARE domain (*Chen et al., 2002*; *Zhou et al., 2015*; *Zhou et al., 2017*). Nevertheless, this suggests STX1A contributes to the regulation of

spontaneous release through two distinct mechanisms: one is inhibitory through its C-terminal half of the SNARE domain, and the other is facilitatory through the palmitoylation of its TMD.

## Materials and methods

### Animal maintenance and generation of mouse lines

All procedures for animal maintenance and experiments were in accordance with the regulations of and approved by the animal welfare committee of Charité-Universitätsmedizin and the Berlin state government Agency for Health and Social Services under license number T0220/09. The STX1-null mouse was generated by breeding the conventional *STX1A* knock-out (KO) line in which exon 2 and 3 are deleted (*Gerber et al., 2008*) with the *STX1B* conditional KO line in which exon 2–4 are flanked by loxP sites (*Wu et al., 2015*). Expression of *Cre* recombinase leads to full STX1-null cells (*Vardar et al., 2016*).

### Neuronal cultures and lentiviral constructs

Hippocampal neurons were obtained from mice of either sex at postnatal day (P) 0–2 and seeded on the preprepared continental or microisland astrocyte cultures as previously described (*Vardar et al., 2016*; *Xue et al., 2007*). The neuronal cultures were then incubated for 13–20 DIV in Neurobasal-A supplemented with B-27 (Invitrogen), 50 IU/ml penicillin and 50 µg/ml streptomycin at 37°C before experimental procedures. Neuronal cultures were transduced with lentiviral particles at DIV 1–3. Lentiviral particles were provided by the Viral Core Facility (VCF) of the Charité-Universitätsmedizin, Berlin, and were prepared as previously described (*Vardar et al., 2016*). The cDNAs of mouse STX1A (NM_016801), STX3A (NM_152220), and STX3B (NM_001025307) were cloned in frame after an NLS-GFP-P2A sequence within the FUGW shuttle vector (*Lois et al., 2002*) in which the ubiquitin promoter was replaced by the human synapsin 1 promoter (f(syn)w). The improved *Cre* recombinase (iCre) cDNA was C-terminally fused to NLS-RFP-P2A. SynGCamp6f was generated analogous to synGCamp2 (*Herman et al., 2014*), by fusing GCamp6f (*Chen et al., 2013*) to the C-terminus of synaptophysin and within the f(syn)w shuttle vector (*Grauel et al., 2016*).

### Western Blot

The lysates were obtained from DIV13-16 high-density neuronal cultures cultivated in 35 mm culture dishes. The neurons were lysed in 200 µl lysis buffer containing 50 mm Tris/HCl, pH 7.9, 150 mm NaCl, 5 mm EDTA, 1% Triton X-100, 0.5% sodium deoxycholate, 1% Nonidet P-40, and 1 tablet of Complete Protease Inhibitor (Roche) for 30 min on ice. Equal amounts of solubilized proteins were loaded in 12% SDS-PAGE and subsequently transferred to nitrocellulose membranes. The membranes were subjected to the following primary antibodies overnight at 4°C according to the experiment: mouse monoclonal anti-β-tubulin III (1:5000; Sigma) or mouse monoclonal anti-actin (1:4000) as internal controls, mouse monoclonal anti-STX1A (1:10,000; Synaptic systems), mouse monoclonal anti-SNAP25 (1:10,000; Synaptic systems), and rabbit polyclonal anti-STX3 (1:1000; Synaptic systems). HRP-conjugated goat secondary antibodies (Jackson ImmunoResearch) were applied for 1 hr at room temperature and detected with ECL Plus Western Blotting Detection Reagents (GE Healthcare Biosciences) in Fusion FX7 image and analytics system (Vilber Lourmat).

### Immunocytochemistry

The high-density cultured hippocampal neurons cultivated on 12 mm culture dishes were fixed with 4% paraformaldehyde (PFA) in 0.1 M phosphate-buffered saline, PH 7.4, for 10 min at DIV13-16. The neurons were then permeabilized with 0.1% Tween–20 in PBS (PBST) for 45 min at room temperature (RT) and then blocked with 5% normal goat serum (NGS) in PBST. Primary antibodies, guinea pig polyclonal anti-Bassoon (1:1000; Synaptic systems) and rabbit polyclonal anti-STX3 (1:1000; Synaptic systems),were applied overnight at 4°C. Subsequently, secondary antibodies, rhodamine red donkey anti-guinea pig IgG (1:500; Jackson ImmunoResearch) and A647 donkey anti-rabbit IgG (1:500; Jackson ImmunoResearch) were applied for 1 hr at RT in the dark. The coverslips were mounted on glass slides with Mowiol mounting agent (Sigma-Aldrich). The images were acquired with an Olympus IX81 epifluorescence-microscope with MicroMax 1300YHS camera using MetaMorph software (Molecular Devices). Exposure times of excitations were kept constant for each wavelength

throughout the images obtained from individual cultures. Data were analyzed offline with ImageJ as previously described (*Vardar et al., 2016*). Sample size estimation was done as previously published (*Vardar et al., 2016*).

## ABE method

$2–3 \times 10^6$ neurons were cultivated on 100 mm culture dishes coated with astrocyte culture and transduced at DIV 1 with *Cre* recombinase in combination with STX1A[WT], STX1A[K260E], or STX1A[CVCV]. All STX1A constructs used for ABE method were N-terminally tagged with FLAG epitope.

For the biotinylation of the proteins, the ABE method was applied as previously described (*Brigidi and Bamji, 2013*). Briefly, the lysates were obtained at DIV 13–16 in lysis buffer, pH 7.2, including 50 mM NEM that was freshly dissolved in EtOH at the day of the experiment. After incubation on ice for 30 min, the lysates were centrifuged and the supernatant was transferred into a fresh tube. 50 µl of anti-FLAG magnetic beads (Sigma) were then added into the lysates for overnight incubation on a rotator at 4°C. The next day, the supernatant was discarded and the anti-FLAG magnetic beads were nutated in lysis buffer, pH 7.2, including 10 mM NEM for 10 min on ice. The beads were then washed thrice with lysis buffer, pH 7.2. The beads were then resuspended with 1 ml lysis buffer, pH 7.2 and separated into half. The supernatant was discarded and one sample from each group was treated with 500 µl lysis buffer, pH 7.2 including 1 M HAM and the other sample without HAM. The HAM solution was freshly prepared at the day of the experiment. The HAM cleavage proceeded for 1 hr on a rotator at RT.

After HAM cleavage, the STX1A bound anti-FLAG magnetic beads were washed thrice in lysis buffer, pH 7.2, and once in lysis buffer, pH 6.2. The beads were then resuspended in lysis buffer, pH 6.2, including 1 µM Biotin-BMCC for 1 hr on a rotator at 4°C. Biotin-BMCC covalently binds to the thiol groups on cysteine residues exposed after the HAM driven cleavage of the palmitates. The supernatant was discarded and the beads were washed thrice in lysis buffer, pH 7.2. The beads were then resuspended in 30 µl 1 X PBS and 1 X SDS-PAGE loading buffer. After incubation at 95°C for 5 min, the supernatant was loaded on an SDS-PAGE and subsequently transferred to a nitrocellulose membrane. The membranes were treated with HRP-conjugated streptavidin antibody (ThermoFisher) for 1 hr at RT. The chemiluminescence was detected in Fusion FX7 image and analytics system (Vilber Lourmat) after treatment of the membrane with ECL Plus Western Blotting Detection Reagents (GE Healthcare Biosciences).

## Electrophysiology

Whole cell patch-clamp recordings were performed on glutamatergic autaptic hippocampal neurons at DIV 14–20 at RT with a Multiclamp 700B amplifier and an Axon Digidata 1550B digitizer controlled by Clampex 10.0 software (both from Molecular Devices). The recordings were analyzed offline using Axograph X Version 1.7.5 (Axograph Scientific).

Prior to recordings, the transduction of the neurons was verified by RFP and GFP fluorescence. Membrane capacitance and series resistance were compensated by 70% and only the recordings with a series resistance smaller than 10 MΩ were used for further recordings. Data were sampled at 10 kHz and filtered by low-pass Bessel filter at 3 kHz. The standard extracellular solution was applied with a fast perfusion system (1–2 ml/min) and contained the following: 140 mM NaCl, 2.4 mM KCl, 10 mM HEPES, 10 mM glucose, 2 mM $CaCl_2$, and 4 mM $MgCl_2$ (300 mOsm; pH 7.4). Borosilicate glass patch pipettes were pulled with a multistep puller, yielding a final tip resistance of 2–5 MΩ when filled with KCl based intracellular solution containing the following: 136 mM KCl, 17.8 mM HEPES, 1 mM EGTA, 4.6 mM $MgCl_2$, 4 mM ATP-Na, 0.3 mM GTP-Na, 12 mM creatine phosphate and 50 U/ml phosphocreatine kinase (300 mOsm; pH 7.4).

The neurons were clamped at –70 mV in steady state. To evoke EPSCs, the neurons were depolarized to 0 mV for 2 ms. The size of the RRP was determined by a 5 s application of 500 mM sucrose in standard external solution (*Rosenmund and Stevens, 1996*) and the total charge transfer was calculated as the integral of the transient current. Fusogenicity measurement was conducted by application of 250 mM sucrose solution for 10 s and calculation of the ratio of the charge transfer of the transient current over RRP. Spontaneous release was determined by monitoring mEPSCs for 30–60 s at –70 mV. To correct false positive events, mEPSCs were recorded in the presence of 3 µM AMPA receptor antagonist NBQX (Tocris Bioscience) in standard external solution.

Sample size estimation was done as previously published (*Rosenmund and Stevens, 1996*).

### SynGcamp6f-imaging

Imaging experiments were performed at DIV 13–16 on autapses in response to a single stimulus and trains of stimuli at 10 Hz as described previously for SynGcamp2-imaging (*Herman et al., 2014*). Images were acquired using a 490 nm LED system (pE2; CoolLED) at a 5 Hz sampling rate with 25ms of exposure time. The acquired images were analyzed offline using ImageJ (National Institute of Health), Axograph X (Axograph), and Prism 8 (Graph-Pad; San Diego, CA). Sample size estimation was done as previously published (*Herman et al., 2014*).

### Statistical analysis

Data in violin graphs present single observations (points), median and the quartiles (lines). Data in x-y plots present means ± SEM. All data were tested for normality with Kolmogorov-Smirnov test. Data from two groups with normal or non-parametric distribution were subjected to Student's two-tailed *t*-test or Mann-Whitney non-parametric test, respectively. Data from more than two groups were subjected to Kruskal-Wallis followed by Dunn's post hoc test when at least one group showed a non-parametric distribution. For data in which all the groups showed a parametric distribution, one-way ANOVA test followed by Tukey's post hoc test was applied. All the tests were run with GraphPad Prism 8.3 and all the statistical data are summarized in corresponding source data tables.

## Acknowledgements

We thank the Charité Viral Core facility, Katja Pötschke, and Bettina Brokowski for virus production, Berit Söhl-Kielczynski and Heike Lerch for technical assistance and to all the Rosenmund Lab members for the discussions. This work was supported by the German Research Council (DFG) grants 388271549, 399894546, 436260754, 278001972.

## Additional information

### Funding

| Funder | Grant reference number | Author |
| --- | --- | --- |
| Deutsche Forschungsgemeinschaft | 388271549 | Christian Rosenmund |
| Deutsche Forschungsgemeinschaft | 399894546 | Christian Rosenmund |
| Deutsche Forschungsgemeinschaft | 436260754 | Christian Rosenmund |
| Deutsche Forschungsgemeinschaft | 27800197 | Christian Rosenmund |

The funders had no role in study design, data collection and interpretation, or the decision to submit the work for publication.

### Author contributions

Gülçin Vardar, Conceptualization, Data curation, Formal analysis, Investigation, Methodology, Validation, Visualization, Writing - original draft, Writing – review and editing; Andrea Salazar-Lázaro, Sina Zobel, Data curation, Formal analysis; Thorsten Trimbuch, Resources; Christian Rosenmund, Conceptualization, Funding acquisition, Project administration, Writing – review and editing

### Author ORCIDs

Gülçin Vardar http://orcid.org/0000-0002-5295-1591
Christian Rosenmund http://orcid.org/0000-0002-3905-2444

## Ethics

All procedures for animal maintenance and experiments were in accordance with the regulations of and approved by the animal welfare committee of Charité;-Universitä;tsmedizin and the Berlin state government Agency for Health and Social Services under license number T0220/09.

## Decision letter and Author response

Decision letter https://doi.org/10.7554/eLife.78182.sa1
Author response https://doi.org/10.7554/eLife.78182.sa2

---

# Additional files

## Supplementary files

• Transparent reporting form

## Data availability

All data generated or analysed during this study are included in the manuscript and supporting file; source data files that contain numerical data used to generate the figures have been provided for all figures. Source data that contain the whole western blot images are provided Figures 3, 4, 6, and 8.

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
