## [Editor Report]

Exocytosis of synaptic vesicles is mediated by synaptic SNARE proteins that overcome the energy barrier for membrane fusion by assembling into a helical bundle, thus pulling the membranes together. Here the authors have used primary cultures of hippocampal neurons obtained from animals in which the isoforms of syntaxin 1, one of the neuronal SNAREs, are deleted, allowing for the introduction of syntaxin 1a mutants in a clean genetic background. Specifically, the authors investigated mutations into the membrane-proximal region and transmembrane domain of syntaxin 1a and they show that not only charge reversal but also mutations preventing palmitoylation of the transmembrane domain have a strong influence on both spontaneous and evoked neurotransmitter release. The results add important details to our mechanistic understanding of the late steps in SNARE-mediated exocytosis.

---

## [Decision Letter]

**Decision letter after peer review:**

Thank you for submitting your article "Syntaxin-1A modulates vesicle fusion in mammalian neurons via juxtamembrane domain dependent palmitoylation of its transmembrane domain" for consideration by *eLife*. Your article has been reviewed by 3 peer reviewers, one of whom is a member of our Board of Reviewing Editors, and the evaluation has been overseen by Richard Aldrich as the Senior Editor. The reviewers have opted to remain anonymous.

Essential revisions:

All referees were positive about the work and state that it constitutes a significant contribution to the understanding of neurotransmitter release, with the experiments being of consistently high quality. While the reviewers do not see the need for additional experiments, they agree that the manuscript should be carefully re-edited to improve clarity and to make the work better understandable to the reader. Moreover, they comment on specific aspects of your work which require clarification, which are detailed below in the statements of the individual referees.

*Reviewer #1 (Recommendations for the authors):*

The manuscript contains an enormous amount of experiments that all are carefully analysed and interpreted. Admittedly, considering the complexity of the results, it is "hard reading" even for someone familiar with the field, and many of the effects are not easy to integrate into a coherent picture. The authors deserve credit for being very considerate and careful in the interpretation of the data. While not breaking new ground, the data further refine our understanding of syntaxin in neuronal exocytosis.

*Reviewer #2 (Recommendations for the authors):*

Overall, this is a well-designed and systematic study that addresses the critical roles of JMD and TMD of syntaxin in neurotransmission. The authors identify a specific role for palmitoylation of syntaxin TMD in regulation of spontaneous release. The results generated by employing gain-of-function mutants are very convincing. Moreover, identification of K260 residues in the JMD for the regulation of palmitoylation of C271/C272 residues in the TMD of syntaxin 1A is a novel finding. However, some points require further clarification.

1. The study is reported in a rather descriptive manner, therefore, the underlying mechanisms are not always clear. For example, K260E and K260A vs. K260Q and K260R (Figure 6) shows that the PIP2 binding is not related to the role of the K260 residue of syntaxin 1A. I think this point would benefit from further clarification.

2. The rationale behind the selection of the GSG insertion (Figure 1) needs to be clarified. Why only three residues? And why the specific GSG residues?

3. The authors should acknowledge that spontaneous – near resting membrane potential – openings of voltage gated ca^2+^ channels may be differentially affected by the mutations and such openings are not measured in this study.

4. A diagram and/or a cartoon summarizing the findings (pointing to the regions of syntaxin and their impact on neurotransmission) would help readers. This would also help alleviate the concern raised in #1 above.

*Reviewer #3 (Recommendations for the authors):*

This manuscript encompasses a comprehensive set of experiments, using state-of-the-art techniques to provide new insight into the molecular context of synaptic vesicle exocytosis. In particular, it is encouraging that Vardar et al. succeed in drawing mechanistic parallels to the mode of action of vesicular SNARE proteins. The results confirm and significantly extend our idea that not only protein-protein but to a large extent also protein-lipid interactions are central for ca^2+^-driven exocytosis and the actual membrane fusion process.

I have no major objections to the conduct of the experiments and the interpretation of the data. Overall, the presentation of the data is clear and the conclusions are valid.

---

## [Author Response]

Reviewer #1 (Recommendations for the authors):The manuscript contains an enormous amount of experiments that all are carefully analysed and interpreted. Admittedly, considering the complexity of the results, it is "hard reading" even for someone familiar with the field, and many of the effects are not easy to integrate into a coherent picture. The authors deserve credit for being very considerate and careful in the interpretation of the data. While not breaking new ground, the data further refine our understanding of syntaxin in neuronal exocytosis.

We thank to reviewer for their time and insights to evaluate our paper. We revised our manuscript in the light of their comments.

Reviewer #2 (Recommendations for the authors):Overall, this is a well-designed and systematic study that addresses the critical roles of JMD and TMD of syntaxin in neurotransmission. The authors identify a specific role for palmitoylation of syntaxin TMD in regulation of spontaneous release. The results generated by employing gain-of-function mutants are very convincing. Moreover, identification of K260 residues in the JMD for the regulation of palmitoylation of C271/C272 residues in the TMD of syntaxin 1A is a novel finding. However, some points require further clarification.

We thank to the reviewer for their time and invaluable comments for the evaluation of our paper. We are pleased to hear that the reviewer found our data convincing. We revised our manuscript in the light of their comments.

1. The study is reported in a rather descriptive manner, therefore, the underlying mechanisms are not always clear. For example, K260E and K260A vs. K260Q and K260R (Figure 6) shows that the PIP2 binding is not related to the role of the K260 residue of syntaxin 1A. I think this point would benefit from further clarification.

We agree with the reviewer that the role of K260 in STX1A’s function is not related to PIP2 binding. However, we argue that the differential phenotypes of K260 mutants is not sufficient to show the PIP2 binding is not related to K260’s function. Firstly, all K260 mutants except K260R showed a reduction in vesicle fusion as well as vesicle priming either significantly or with a strong tendency (Figure 6). It is well characterized that a PIP2 binding motif is flanked by a single lysine or arginine residue on one end and double lysine or arginine residues at the other end. Therefore, the rescue of neurotransmitter release parameters by K260R mutant might well be due to the restoration of PIP2-STX1A binding. Therefore, the differential phenotypes of K260 mutants do not exclude the PIP2 binding as the main function of STX1A’s K260 residue.

On the other hand, we now included the analysis of double charge neutralization mutations of STX1A’s JMD as a read-out of the effects of the inhibition of PIP2-STX1A interaction, even though this interaction still remains controversial. Similar to previous results, we observed that these mutants have more dramatic effect in vesicle fusion and priming. As the single charge reversal mutants have milder phenotypes compared to multiple charge neutralization mutations, we propose that STX1A’s JMD has a function in vesicle fusion downstream of PIP2 binding. We have included this interpretation into our discussion in lines 431-442.

2. The rationale behind the selection of the GSG insertion (Figure 1) needs to be clarified. Why only three residues? And why the specific GSG residues?

We are sorry if we were not clear enough with our rationale behind our mutational designs. It is well known that inserting GSG residues into a helical structure elongates its length by one helical turn and thus less than 1 nm without altering the gross structure of the protein. Therefore, we used only three amino acids to elongate the JMD of STX1A to reach a minuscule change in the intermembrane distance along the SNARE complex without largely affecting the assembly of the vesicular release machinery. Furthermore, previous studies utilized a similar approach for similar questions and we preferred to employ similar mutations to be able to better interpret our data in the context of known data. We now have clarified this issue in lines 71-77.

3. The authors should acknowledge that spontaneous – near resting membrane potential – openings of voltage gated ca^2+^ channels may be differentially affected by the mutations and such openings are not measured in this study.

We agree with the reviewer that our data do not address single ca^2+^-channel opening and the inhibition of the stochastic opening of ca^2+^-channels might well be the reason of reduced spontaneous neurotransmitter release. However, out AP-driven ca^2+^-influx assay show that the STX1-null neurons that exogenously express STX3A in any form have a dramatically lower ca^2+^-entry into the synapse compared to STX1A neurons (Figure 9). This suggests a large decrease in the ca^2+^-channel abundance and/or activity. As mutant STX3A that has two cysteine residues in its TMD combined with STX1A’s JMD or that has the entire JMD and TMD of STX1A leads to 3-4 fold increase in spontaneous release (Figure 8), we propose that the JMD-TMD has an additional role in the facilitation of vesicle fusion besides its putative interaction with ca^2+^-channels. In this light, we hypothesize that JMD dependent palmitoylation of STX1A’s TMD increases the number of SVs that more easily overcome the energy barrier for membrane merger.

We have discussed this issue more in detail in our discussion lines 516-535.

4. A diagram and/or a cartoon summarizing the findings (pointing to the regions of syntaxin and their impact on neurotransmission) would help readers. This would also help alleviate the concern raised in #1 above.

We have included a speculative model for the effects of K260, C271, and C272 in STX1A on neurotransmitter release and the energy barrier for membrane merger (Figure 6).

Reviewer #3 (Recommendations for the authors):This manuscript encompasses a comprehensive set of experiments, using state-of-the-art techniques to provide new insight into the molecular context of synaptic vesicle exocytosis. In particular, it is encouraging that Vardar et al. succeed in drawing mechanistic parallels to the mode of action of vesicular SNARE proteins. The results confirm and significantly extend our idea that not only protein-protein but to a large extent also protein-lipid interactions are central for ca^2+^-driven exocytosis and the actual membrane fusion process.I have no major objections to the conduct of the experiments and the interpretation of the data. Overall, the presentation of the data is clear and the conclusions are valid.

We thank to the reviewer for their time and insights to evaluate our manuscript. We are pleased to hear that the reviewer found our data convincing.